# Chromatin accessibility mapping of the striatum identifies tyrosine kinase FYN as a therapeutic target for heroin use disorder

Gabor Egervari[1,2,8], Diana Akpoyibo[1,2,3], Tanni Rahman [1,2,3], John F. Fullard[1,4,5], James E. Callens[1,2,3], Joseph A. Landry[1,2,3], Annie Ly[1,2,3], Xianxiao Zhou [4,5], Noël Warren[1,2,3], Mads E. Hauberg [1,4,5], Gabriel Hoffman [4,5], Randy Ellis [1,2,3], Jacqueline-Marie N. Ferland[1,2,3], Michael L. Miller[1,2], Eva Keller[6], Bin Zhang[4,5], Panos Roussos [1,4,5,7] & Yasmin L. Hurd [1,2,3 ✉]

The current opioid epidemic necessitates a better understanding of human addiction neurobiology to develop efficacious treatment approaches. Here, we perform genome-wide assessment of chromatin accessibility of the human striatum in heroin users and matched controls. Our study reveals distinct neuronal and non-neuronal epigenetic signatures, and identifies a locus in the proximity of the gene encoding tyrosine kinase *FYN* as the most affected region in neurons. *FYN* expression, kinase activity and the phosphorylation of its target Tau are increased by heroin use in the post-mortem human striatum, as well as in rats trained to self-administer heroin and primary striatal neurons treated with chronic morphine in vitro. Pharmacological or genetic manipulation of FYN activity significantly attenuates heroin self-administration and responding for drug-paired cues in rodents. Our findings suggest that striatal FYN is an important driver of heroin-related neurodegenerative-like pathology and drug-taking behavior, making FYN a promising therapeutic target for heroin use disorder.

[1] Department of Psychiatry, Icahn School of Medicine at Mount Sinai, New York, NY 10029, USA. [2] Department of Neuroscience, Friedman Brain Institute, Icahn School of Medicine at Mount Sinai, New York, NY 10029, USA. [3] Addiction Institute, Icahn School of Medicine at Mount Sinai, New York, NY 10029, USA. [4] Department of Genetics and Genomic Science, Icahn School of Medicine at Mount Sinai, New York, NY 10029, USA. [5] Institute for Multiscale Biology, Icahn School of Medicine at Mount Sinai, New York, NY 10029, USA. [6] Department of Forensic and Insurance Medicine, Semmelweis University, Budapest, Hungary. [7] Mental Illness Research, Education, and Clinical Center (VISN 2 South), James J. Peters VA Medical Center, Bronx, NY 10468, USA. [8]Present address: Department of Cell and Developmental Biology, University of Pennsylvania, Philadelphia, PA 19104, USA. ✉email: Yasmin.Hurd@mssm.edu

Heroin-use disorder continues to impose tremendous suffering and financial costs on society. Therefore, a better understanding of neuronal maladaptations contributing to this disorder is imperative to drive medication development. Historically, the field has focused on animal models of addiction, and surprisingly little direct molecular information is available about the addicted human brain[1]. Therefore, further studies of post-mortem human brains[2] are critical to identify molecular alterations in long-term heroin users compared to age- and gender-matched controls.

Epigenetic mechanisms are dynamic processes that regulate gene expression and have recently emerged as key contributors to the molecular impairments caused by exposure to environmental factors such as abused substances[3–5]. We previously showed that chronic heroin use leads to hyperacetylation of specific lysine residues of histone H3 (H3K27ac) at specific genomic loci[2]. These alterations contribute to transcriptional changes related to glutamatergic neurotransmission and have a significant role in mediating addiction-like behaviors[2]. Nonetheless, epigenetic regulation is extremely complex and studying a single post-translational histone modification provides limited insight into global chromatin alterations. Rather, transcriptional activity is influenced by an elaborate combination of epigenetic marks, and direct assessment of chromatin state may afford a less biased and a more informative measure of gene regulation.

Here, we study dorsal striatal chromatin accessibility directly using the assay for transposase accessible chromatin coupled with high-throughput sequencing (ATAC-seq)[6]. We focus on the striatum due to its essential role in regulating reward, inhibitory control, motivation and goal-directed behaviors, impairments of which are linked to substance use disorders[7]. Using an established protocol that combines FANS (fluorescence-assisted nuclei sorting) and ATAC-seq, we generate quantitative open chromatin profiles in neuronal and non-neuronal nuclei[8,9] and compare genome-wide chromatin accessibility between heroin users and matched controls. We identify a region near the gene encoding for tyrosine kinase *FYN* as most significantly affected by heroin in neurons. We show that this locus is a putative regulatory element that enhances transcriptional activity in vitro. In vivo, *FYN* expression and activity are significantly induced by heroin. Consistent with FYN kinase's implication in tauopathy disorders and substance abuse[10], we detect elevated phosphorylation of Tau at a residue directly targeted by FYN. Strikingly, saracatinib, a FYN inhibitor currently in clinical trials for Alzheimer's disease[11], attenuates addiction-like behaviors in rats self-administering heroin, similar to effects observed with siRNA-mediated knockdown of *Fyn* in the dorsal striatum. Overall, our findings suggest that opioid exposure induces neurodegenerative cellular processes associated with epigenetic disturbances and that the use of FYN inhibitors could be promising for heroin medication development.

## Results

**Heroin affects chromatin accessibility in the striatum**. We screened heroin-related alterations in chromatin accessibility across the genome by performing ATAC-seq in the post-mortem dorsal striatum (putamen) of long-term heroin users ($n = 10$) and matched controls ($n = 10$; Supplementary Data 1; Fig. 1a). Considering the high cellular heterogeneity of the human striatum, we carried out ATAC-seq on FANS-sorted neuronal and non-neuronal (mainly glial) cells based on the presence or absence of nuclear marker NeuN (feminizing locus on X-3), a nuclear protein specific to neurons in vertebrates[12]. We were able to generate high quality libraries (Supplementary Data 2) meeting or exceeding guidelines set by ENCODE. Importantly, we observed high ATAC signal at the transcription start site (TSS) of neuronal

genes in neuronal (blue) but not in non-neuronal (green) cells (e.g., *CAMK2A*; Fig. 1a). Similarly, we observed high ATAC signal at the TSS of glial genes in non-neuronal but not in neuronal cells (e.g., *OLIG2*; Fig. 1a). Furthermore, we compared our ATAC-seq data to publicly available RNA-seq data from the CommonMind Consortium and found that, as predicted, "open genes" [as defined by the presence of ATAC peak(s) within 10 kb of the TSS] had significantly higher gene expression compared to "closed genes" (Supplementary Fig. 1).

We identified a total of 106,247 and 67,750 accessible regions in neuronal and non-neuronal samples, respectively (Supplementary Data 3 and 4). The majority of ATAC-seq peaks mapped to non-coding regulatory elements within introns and intergenic regions (Supplementary Fig. 2A–D) with substantial overlaps between heroin users and controls related to homeostatic genes (Supplementary Fig. 2E, F). Using genome feature enrichment/depletion analysis[13], we found that neuronal peaks from heroin users were specifically enriched in CpG islands, promoters and 5′UTRs, whereas centromeres, introns, gaps and intergenic areas were depleted across all groups (Fig. 1b). Strikingly, while repressed segments constituted the top enriched regulatory category in neurons from control subjects and non-neuronal cells from both heroin and control subjects, neuronal peaks from heroin users uniquely overlapped with both enhancer regions marked by H3K4me1 (Fisher's exact test, corrected $p = 1.58 \times 10^{-179}$) and EZH2 binding sites (Fisher's exact test, corrected $p = 4.97 \times 10^{-123}$). Although no conclusive statements can be made without cell-type-specific transcriptomic analysis, these differences suggest that overall transcriptional activity may be affected in neurons of heroin users (Fig. 1c; Supplementary Data 5–12). This was further supported by a significant increase in promoter peaks in neurons from heroin users compared to controls (chi-square test, $\chi^2 = 837.79$, df = 1, $p = 3.28 \times 10^{-184}$, Supplementary Fig. 2A, B).

We next examined the proportion of variance in open chromatin regions explained by cell type, heroin-use status (i.e., heroin or control), gender and post-mortem interval (PMI). This analysis is of particular interest for smaller post-mortem human cohorts, where criteria for genome-wide significance are rarely met. Most of the variance in chromatin accessibility (average 38.9%) was explained by cell type (Fig. 2a) and, accordingly, principal component analysis (PCA) revealed good separation of neuronal and non-neuronal samples (Supplementary Fig. 3). Other major contributors included individual variability (average 7.4%), gender (average 3.2%), and PMI (average 2.7%). As expected, the majority of open chromatin with variance >50% explained by gender were found on sex chromosomes with 167 on chromosome Y and 93 on chromosome X out of 271 regions. Because the majority of variance was attributed to cell type (Fig. 2a), we then modeled drug group variance separately for each cell type. In this analysis, heroin-use status explained 7.5% and 4.9% of the average variance in neuronal and non-neuronal peaks, respectively. Thus, strikingly, this contribution exceeded that of individual variability, gender or PMI. Taken together, these data indicate that long-term heroin use profoundly affects striatal chromatin accessibility in a cell-type-specific manner.

We subsequently performed differential chromatin accessibility analysis for heroin-use status in neuronal and non-neuronal peaks and observed strong quantitative differences between these groups using a threshold of nominal $p < 0.01$ and $|logFC| > 1.7$ (Fig. 2b, c). Gene ontology analysis revealed that regions of less accessible chromatin in neuronal nuclei from heroin users (Fig. 2d) enriched for plasma membrane components (minimum hypergeometric test (mHG), adjusted $p = 4.03 \times 10^{-6}$), transmembrane receptor activity (mHG, $p = 5.6 \times 10^{-3}$), transcription factor activity (mHG, $p = 5.6 \times 10^{-3}$), RNApolII activity (mHG, $p = 1.59 \times 10^{-3}$), as well as regulation of nervous system development (mHG, $p = 2.14 \times 10^{-6}$),

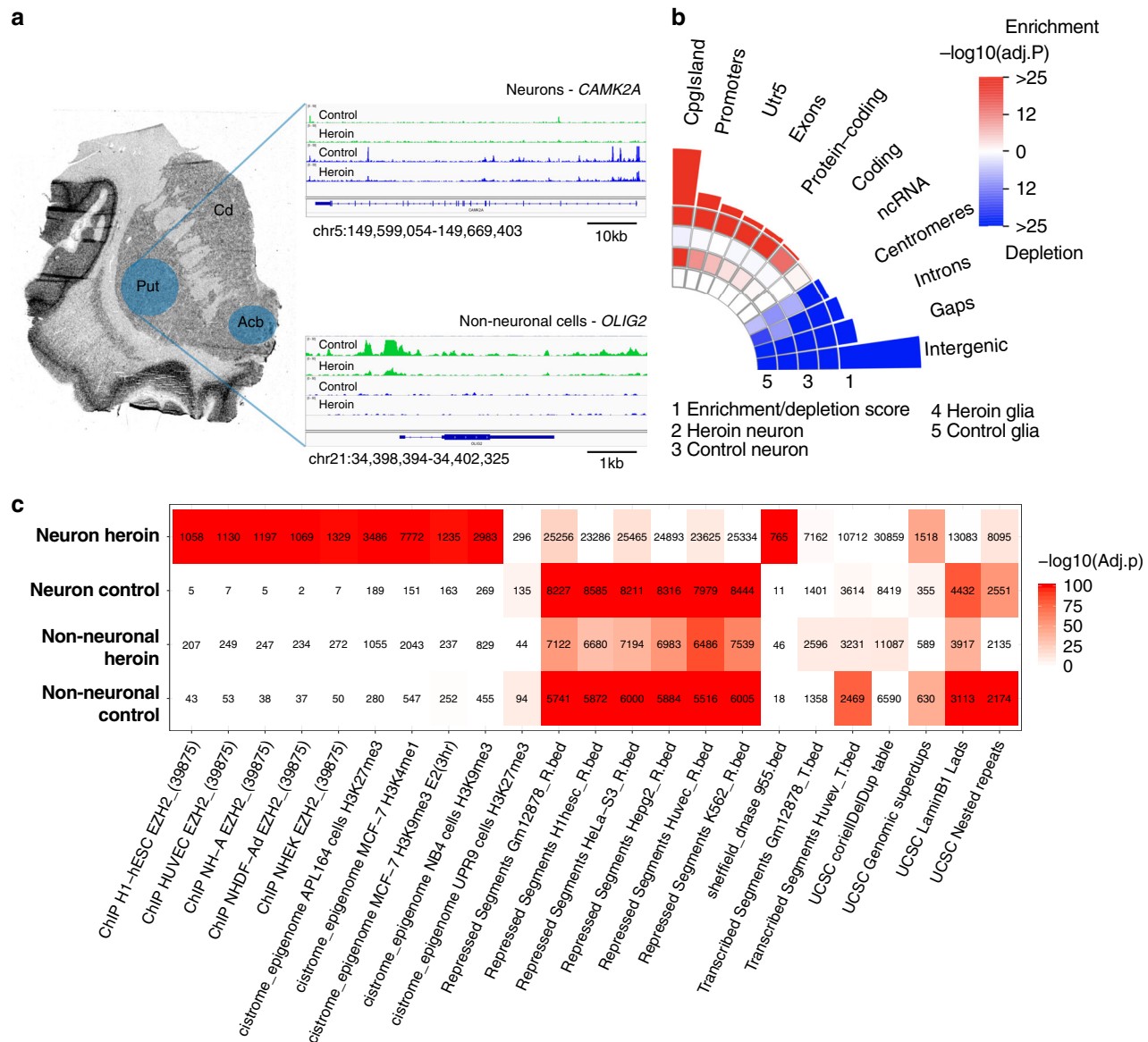

**Fig. 1 Heroin exposure increases chromatin accessibility of neuronal promoters. a** Schematic of the experimental design. Putamen and nucleus accumbens tissue (blue shades) was collected from post-mortem brains of human heroin users and controls. ATAC-seq was performed in FANS-sorted neuronal and non-neuronal (predominantly glial) cell populations (n = 10/group). Neuronal cells (blue tracks) showed accessible regions at the *CAMK2A* gene but not at glial marker *OLIG2*. Conversely, the *OLIG2* locus but not the *CAMK2A* locus was accessible in non-neuronal cells (green tracks). **b** Neuronal ATAC peaks from heroin users were specifically enriched in CpG islands, promoters and 5′UTRs, whereas centromeres, introns, gaps, and intergenic areas were depleted across all groups (Fisher's exact test, Benjamini–Hochberg correction). **c** Heat map showing the most enriched regulatory elements from ENCODE in neurons and non-neuronal cells from heroin users and control subjects. Neuronal ATAC peaks from heroin users were enriched for enhancer regions marked by H3K4me1 and EZH2 binding sites. Acb nucleus accumbens, Put putamen, Cd nucleus caudatus, utr untranslated region, ncRNA non-coding RNA.

neurogenesis (mHG, $p = 4.61 \times 10^{-6}$) and morphogenesis of branching (mHG, $p = 7.59 \times 10^{-3}$). Regions that were more accessible in heroin-sourced neuronal nuclei compared to control (Fig. 2e) were enriched for genes related to synaptic membrane (mHG, $p = 5.74 \times 10^{-7}$), neuronal projections (mHG, $p = 3.24 \times 10^{-4}$), post-synaptic density (mHG, $p = 1.3 \times 10^{-3}$), dendrite (mHG, $p = 2.89 \times 10^{-2}$), glutamate receptor activity (mHG, $p = 4.5 \times 10^{-3}$), as well as regulation of nervous system development (mHG, $p = 8.81 \times 10^{-4}$), modulation of synaptic transmission (mHG, $p = 3.7 \times 10^{-3}$) and regulation of synaptic plasticity (mHG, $p = 6.48 \times 10^{-3}$). Emphasizing the specificity of these findings, matching GO analyses from non-neuronal cell populations highlighted enrichment for categories related to cell adhesion,

signal transduction, and nucleotide exchange (Supplementary Fig. 4).

Intriguingly, heroin had markedly different effects on chromatin accessibility in neurons compared to non-neuronal cells, especially with respect to open chromatin (Fig. 2f, g). Using a cutoff of nominal $p < 0.01$ for the heroin/control comparison, many more regions were significantly affected by heroin in neurons (Fig. 2f) compared to non-neuronal cells (Fig. 2g). Moreover, affected non-neuronal regions largely overlapped with affected neuronal regions (red dots in Fig. 2g), whereas neurons had more unique significantly affected regions (gray dots in Fig. 2f). Interestingly, these differences were more pronounced for more open regions, suggesting that heroin might activate specific

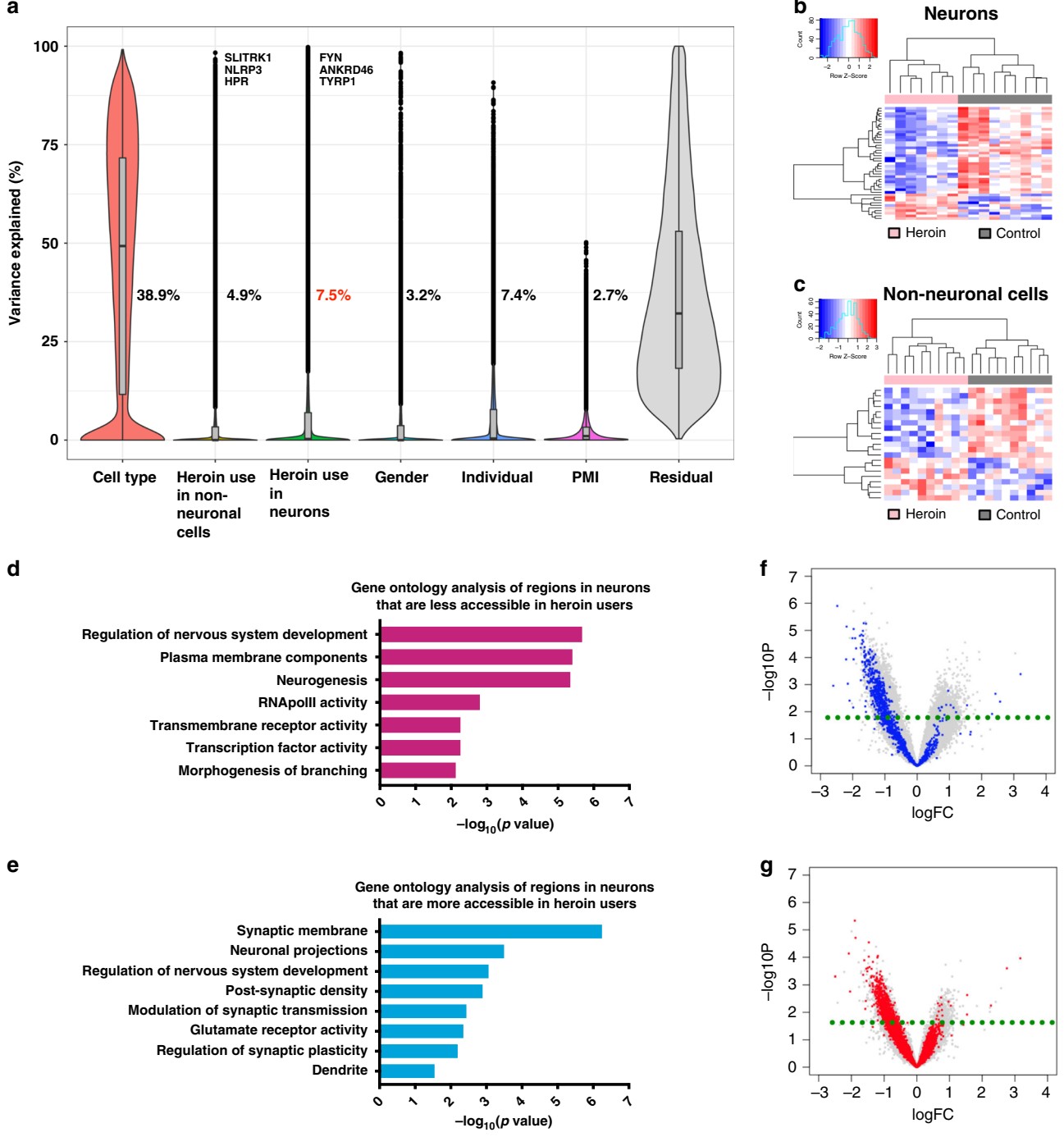

**Fig. 2 Genome-wide changes of chromatin state in the post-mortem striatum. a** Factors contributing to ATAC-seq signal variability. Violin plots showing the percentage of ATAC-seq signal variance explained by cell type (neuronal or non-neuronal), diagnosis (heroin vs. control; shown separately in neuron and non-neuronal cells), gender, individual variability, post-mortem interval, and residual factors. Top genes are indicated next to the plots. Within neurons, heroin-use group (heroin vs. control) was the primary source of variability. In this cell population, a peak close to the *FYN* gene ranked first as the signal whose variance was most influenced by drug group. PMI: post-mortem interval. Boxplots indicate median, 25th and 75 quantiles. The whiskers are definted as 1.5× inter-quartile range (IQR). **b**, **c** Heat maps of differentially accessible regions in neuronal (**b**) and non-neuronal (**c**) cell populations revealed marked quantitative group differences. **d** Enriched gene ontology categories for neuronal ATAC-seq peaks that are less accessible in heroin users (minimum hypergeometric test). **e** Enriched gene ontology categories for neuronal ATAC-seq peaks that are more accessible in heroin users include terms related to synaptic function and membrane receptor activity (minimum hypergeometric test). **f**, **g** Volcano plots showing differentially accessible regions in heroin users compared to controls in neuronal (**f**) and non-neuronal (**g**) cell populations. Significant non-neuronal peaks are marked in blue on the neuronal plot (**f**) and significant neuronal peaks are marked in red on the non-neuronal plot (**g**). More regions were significantly affected by heroin in neurons, whereas regions affected in glia were mostly shared between the two cell populations. PMI post-mortem interval.

transcriptional programs selectively in neurons. Next, to determine DNA binding factor occupancy genome wide, we performed footprinting analysis of our neuronal ATAC-seq data[6]. We found significant enrichment of several transcription factors and DNA binding proteins including, e.g., DNMT1 and SMAD1 in heroin users compared to controls (Supplementary Fig. 5). Of note, many were previously implicated in neuronal function, behavior and addiction. These included, for example, MBD2[14,15] and MeCP2[16,17].

Taken together, the genome-wide ATAC-seq data identified alterations in neuronal chromatin at genes related to synaptic activity in the human dorsal striatum (putamen), which might contribute to the transcriptional regulation of long-term adaptations underlying heroin-use disorder.

**Heroin induces the expression and activity of FYN kinase.** Given that open chromatin regions show substantial variability in heroin versus control neurons, we next ranked neuronal peaks based on the contribution of heroin use to variance in peak amplitude (Fig. 2a). We prioritized this analysis as strict criteria for genome-wide significance are rarely met in molecular studies of the post-mortem human brain[1]. Intriguingly, the unbiased variance analysis identified a peak in the proximity of FYN (chr6:112232297-112232846; ~38 kb upstream of the transcription start site), member of the SRC tyrosine kinase family, as the genomic locus most significantly affected by heroin use (68.4% variance explained by disease, which strongly exceeded the contribution of gender [0%], post-mortem interval [2.4%] and residual variance [29.3%] at this locus; Fig. 3a). This is especially intriguing considering that FYN is a member of the glutamatergic post-synaptic density which has a central role in the regulation of drug-seeking behavior and relapse[2,18–21]. To investigate whether the region identified by the variance analysis is a potential regulatory element, we first assessed the enrichment of specific histone marks in human brain samples from ENCODE (Supplementary Data 13). Strikingly, we found significant enrichment for enhancer marks H3K4me1 and H3K27ac in this region (Fig. 3a). Furthermore, to identify potential higher order chromatin interactions between our region of interest and gene promoters, we queried a published PLAC-seq (proximity ligation assisted chromatin immunoprecipitation) dataset[22]. We found that in neurons, oligodendrocytes and microglia, this locus interacts with both upstream and downstream gene promoters, including that of FYN (Supplementary Fig. 6). Together, the histone modification profile of this region and local 3D interactions suggest that it might act as a regulatory element. To further validate this locus as a putative enhancer, we amplified this region from human genomic DNA and cloned it into a pGL3 promoter vector driven by SV40 promoter (Fig. 3b). We then transfected HEK293 cells with hFyn-pGL3 or pGL3 control vector along with a reporter vector expressing renilla luciferase under the SV40 promoter as internal control of transfection efficiency. Using a Dual-Glo luciferase assay, we demonstrated that the putative upstream regulatory region significantly increased luciferase activity indicative of enhancing gene expression ($n = 3$/group, Student's $t$-test, $t_4 = 4.505$, $p = 0.0108$; Fig. 3c). Taken together, our in silico and in vitro data suggest that the unbiased variance analysis of ATAC signal identified a region of potential functional relevance upstream of FYN, that is strongly affected by chronic heroin use in the post-mortem human brain.

In order to more closely characterize the FYN promoter, we next investigated chromatin accessibility throughout a 600 bp region (chr6:112194332–112194931) overlapping the transcription start site of FYN. Interestingly, the promoter region of FYN was more accessible in neurons of heroin users compared to controls (2.2-fold increase in average ATAC signal over this region, $t = 14.46$, $p = 7.64 \times 10^{-93}$ from two-tailed Student's $t$-test; Fig. 4a), while this difference was less substantial in non-neuronal cells (1.3-fold increase in average ATAC signal over this region, $t = 6.395$, $p = 2.3 \times 10^{-10}$ from two-tailed Student's $t$-test; Fig. 4b). More accessible chromatin in this region suggests enhanced transcriptional activity of FYN. In fact, using qPCR ($n = 30$ heroin users and $n = 16$ controls), we observed elevated FYN mRNA levels in the putamen (homogenate tissue) of human heroin users compared to matched controls (Student's $t$-test, $t_{44} = 1.859$, $p = 0.0349$, Fig. 4c). Furthermore, FYN transcript expression was positively correlated with several post-synaptic density members (GRIA1, GRM5, HOMER1 and DLG4) in heroin users but not in controls (Supplementary Fig. 7) further emphasizing the important role of FYN in heroin use-related post-synaptic density pathology.

In addition, FYN mRNA levels tended to positively correlate with years of previous drug use in our post-mortem population (linear regression, $r = 0.37$, $F_{1,19} = 2.924$, $p = 0.1036$, Supplementary Fig. 8A) and showed a significant negative correlation with urine morphine levels at the time of autopsy (linear regression, $r = -0.40$, $F_{1,22} = 4.403$, $p = 0.0472$, Supplementary Fig. 8B), suggesting antagonistic effects of chronic heroin use and acute morphine toxicity on FYN expression. Intriguingly, these correlations were present on the protein level as well (Supplementary Fig. 8C–E) and Fyn protein increased in a dose-dependent manner in striatal monocultures chronically treated with morphine (Supplementary Fig. 8F). Similar opposing correlations were previously reported for histone H3 acetylation in our post-mortem population[2]. These findings emphasize the dynamic nature of epigenetic and transcriptional dysregulation in relation to the acute and long-term effects of heroin use.

In order to assess the causal relationship between heroin use and FYN pathology, we next sought to determine drug-related changes in Fyn expression using animal models of heroin-use disorder. Specifically, we focused on the anterior, more medial part of the dorsal striatum, which is the area where long-lasting activation of Fyn has been previously described by Ron and colleagues during alcohol use[23–25]. Adult male Long-Evans rats were trained to self-administer heroin and tissue was collected 24 h after the last self-administration session for post-mortem studies. Consistent with the human findings, Fyn mRNA was significantly increased in the dorsal striatum (caudate-putamen) of rats self-administering heroin ($n = 12/28$ control/heroin, Student's $t$-test, $t_{37} = 2.305$, $p = 0.0269$, Fig. 4d) and in primary striatal cultures exposed to morphine in vitro (one-way ANOVA $F_{2,13} = 6.419$, $p = 0.0115$; significant increase observed with 100 μM morphine, $t_{10} = 3.785$, $p = 0.0036$; Fig. 4e).

Importantly, we validated heroin-induced changes in dorsal striatal FYN transcription in several independent, publicly available datasets (Supplementary Data 14). Of note, other SRC family members were less frequently affected. In addition, FYN mRNA was unchanged in the nucleus accumbens of human heroin users ($n = 16/21$ control/heroin, Student's $t$-test, $t_{35} = 0.09115$, p $= 0.9279$; Supplementary Fig. 9A) and heroin self-administering rats ($n = 8/11$ control/heroin, Student's $t$-test, $t_{17} = 1.55$, $p = 0.1395$; Supplementary Fig. 9B), emphasizing the specificity of dorsal striatal findings.

Next, we assessed Fyn kinase activity in the caudate-putamen of heroin self-administering rats directly by immunoprecipitating total Fyn and western blotting with antibodies specific to the active (Y418) or inactive (Y529) phosphorylated forms of Src family kinases[10]. We found that heroin led to significantly increased active p(Y418)-Fyn ($n = 8/12$ control/heroin, Student's $t$-test, $t_{18} = 2.563$, $p = 0.0196$) and significantly decreased inactive p(Y529)-Fyn (Student's $t$-test, $t_{18} = 2.411$, $p = 0.0268$) protein

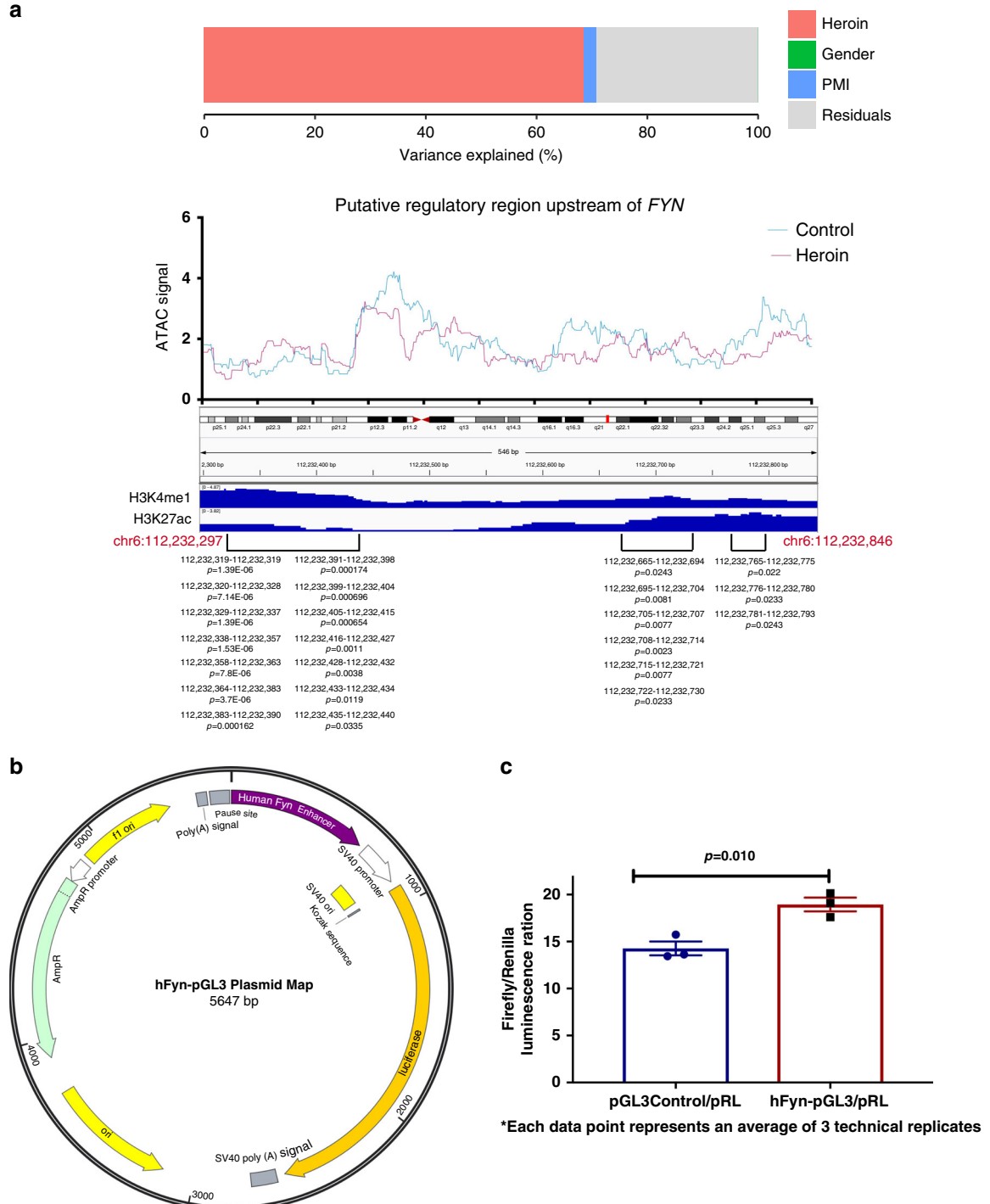

**Fig. 3 Variance analysis of ATAC signal identifies a putative regulatory element upstream of *FYN*. a** Variance distribution, raw ATAC signal, and enrichment of H3K4me1 and H3K27ac enhancer marks (ENCODE, human caudate nucleus; nominal signal *p*-values from ENCODE are shown) over the putative *FYN* regulatory locus. The contribution of heroin use to ATAC signal variance in this locus strongly exceeds that of other factors. H3K4me1 and H3K27ac are significantly enriched in this region, suggesting it might be an active enhancer **b** hFyn-pGL3 reporter plasmid map. **c** Dual-glo luciferase assay in HEK293 cells revealed significantly increased luciferase activity in cells transfected with hFyn-pGL3 compared to pGL3 control vector (*n* = 3/group, Student's *t*-test, $t_4$ = 4.505, *p* = 0.0108). Data are represented as mean ± SEM. Source data are provided as a Source Data file.

levels (Fig. 4f–h). In addition, active (linear regression, $F_{1,18}$ = 5.854, *r* = 0.50, *p* = 0.0263) but not inactive p-Fyn (linear regression, $F_{1,18}$ = 0.4255, *r* = −0.15, *p* = 0.5225) showed a positive correlation with active lever pressing (Fig. 4i, j), further supporting a positive relationship between heroin use and dorsal striatal Fyn kinase activity. Strikingly, increased activity of Fyn

persisted up to 2 weeks following the last self-administration session (*n* = 8/group, Student's *t*-test, $t_{14}$ = 2.403 and *p* = 0.0307 for active p-Fyn and $t_{14}$ = 2.265, *p* = 0.0399 for inactive p-Fyn; Fig. 4k–m). Although similar trends were also observed following a single bolus injection of 1 mg/kg heroin (*n* = 8/group, Student's *t*-test, $t_{14}$ = 1.918 and *p* = 0.0757 for active p-Fyn and $t_{14}$ = 2.032,

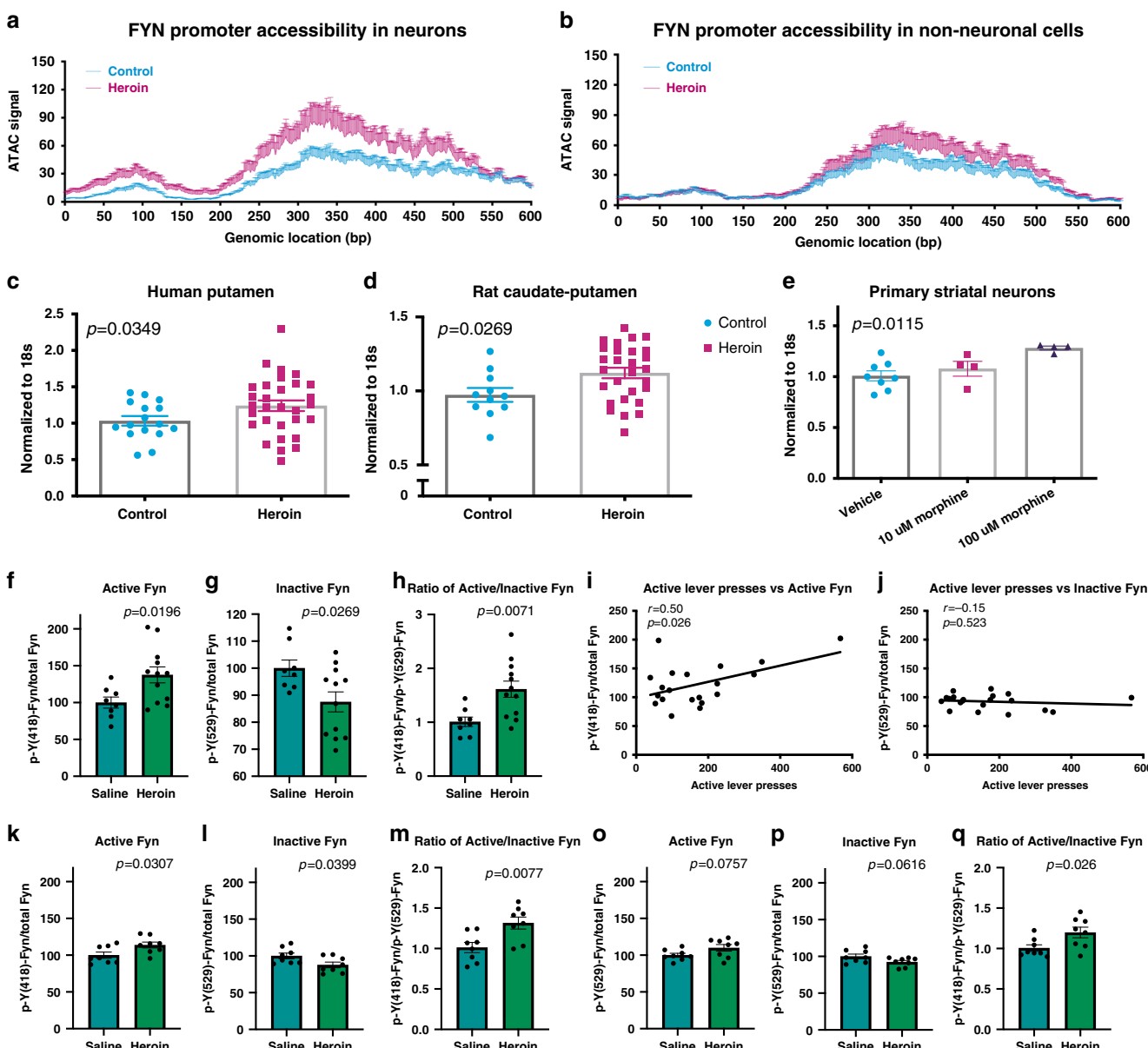

**Fig. 4 Heroin leads to enhanced chromatin accessibility, transcription, and activity of FYN. a**, **b** ATAC-seq signal over a 600 bp region (chr6:112194332–112194931) overlapping the transcription start site of *FYN* in NeuN-positive neuronal (**a**) and NeuN-negative non-neuronal (**b**) cell populations. Accessibility of this region is markedly increased by heroin, especially in neuronal cells. **c** *FYN* mRNA levels were increased in the putamen of human heroin users compared to matched controls ($n = 30$ heroin users, $n = 16$ controls, Student's *t*-test, $t_{44} = 1.859$, $p = 0.0349$). **d** *Fyn* mRNA levels were increased in the caudate-putamen of heroin self-administering rats ($n = 12/28$ control/heroin, Student's *t*-test, $t_{37} = 2.305$, $p = 0.0269$). **e** *Fyn* mRNA levels were increased in primary striatal cells treated with morphine (one-way ANOVA $F_{2,13} = 6.419$, $p = 0.0115$; significant increase observed with 100 μM morphine, Student's *t*-test $t_{10} = 3.785$, $p = 0.0036$). **f** Active p(Y418)-Fyn levels were increased in the dorsal striatum of heroin self-administering rats 24 h after the last session ($n = 8/12$ control/heroin, Student's *t*-test, $t_{18} = 2.563$, $p = 0.0196$). **g** Inactive p(Y529)-Fyn levels were decreased in the same animals ($n = 8/12$ control/heroin, Student's *t*-test, $t_{18} = 2.411$, $p = 0.0268$). **h** Ratio of active/inactive p-Fyn was significantly increased in the same animals ($n = 8/12$ control/heroin, Student's *t*-test, $t_{18} = 3.037$, $p = 0.0071$). **i**, **j** Active (I; linear regression, $F_{1,18} = 5.854$, $r = 0.50$, $p = 0.0263$) but not inactive (J; linear regression, $F_{1,18} = 0.4255$, $r = -0.15$, $p = 0.5225$) dorsal striatal p-Fyn correlated positively with active lever pressing. **k** Active p(Y418)-Fyn levels were increased in the dorsal striatum of heroin self-administering rats 2 weeks after the last session ($n = 8$/group, Student's *t*-test, $t_{14} = 2.403$, $p = 0.0307$). **l** Inactive p(Y529)-Fyn levels were decreased in the same animals ($n = 8$/group, Student's *t*-test, $t_{14} = 2.265$, $p = 0.0399$). **m** Ratio of active/inactive p-Fyn was significantly increased in the same animals ($n = 8$/group, Student's *t*-test, $t_{14} = 3.106$, $p = 0.0077$). **n** Active p(Y418)-Fyn levels showed an increasing trend ($n = 8$/group, Student's *t*-test, $t_{14} = 1.918$, $p = 0.0757$) in the dorsal striatum of naive rats injected with a single dose of 1 mg/kg heroin. **o** Inactive p(Y529)-Fyn levels showed a decreasing trend in the same animals ($n = 8$/group, Student's *t*-test, $t_{14} = 2.032$, $p = 0.0616$). **p** Ratio of active/inactive p-Fyn was increased in the same animals ($n = 8$/group, Student's *t*-test, $t_{14} = 2.492$, $p = 0.0259$). Data are represented as mean ± SEM (normalized to saline group in **f**–**p**). Source data are provided as a Source Data file.

$p = 0.0616$ for inactive p-Fyn; Fig. 4n–p), chronic, repeated exposures to the drug as experienced during heroin self-administration were necessary to induce significant changes.

Taken together, our data indicate that chronic heroin use leads to increased accessibility, gene expression and tyrosine kinase activity of FYN in the dorsal striatum. Of note, we found that other SRC family members were not affected in terms of chromatin accessibility (Supplementary Fig. 10A) or kinase activity ($n = 6/8$ control/heroin; Student's $t$-test, $t_{12} = 1.207$ and $p = 0.2508$ for active p-Src, Student's $t$-test, $t_{12} = 0.6566$ and $p = 0.5238$ for inactive p-Src; Supplementary Fig. 10B–D).

**Heroin leads to Tau phosphorylation in the striatum.** FYN is strongly implicated in Alzheimer's disease and other neurodegenerative tauopathies[26–28]. This kinase directly phosphorylates Tau[27], a microtubule scaffolding protein abundant in neurons that regulates cytoskeleton organization, and its hyper-phoshporylated form leads to the formation of intraneuronal aggregates called neurofibrillary tangles (NFT), the primary neuropathological marker of Alzheimer's disease. As we previously detected increased phosphorylation of Tau in the post-mortem brain of human heroin abusers[29], we hypothesized that heroin-induced increases in FYN might be driving Tau phosphorylation in these individuals, especially at Tyrosine-18, a residue specifically targeted by the FYN kinase[27]. We first tested this hypothesis in vitro by treating primary striatal-cortical co-cultures with morphine (10 μM every other day for 6 days) and observed a significant increase of Tyrosine-18 phosphorylated Tau ($n = 3$/group, Student's $t$-test, $t_4 = 4.374$, $p = 0.0119$; Fig. 5a). Moreover, we found significant increases in Tyrosine-18 phosphorylation of Tau in heroin self-administering rats, both 1 h ($n = 5$/group, Student's $t$-test, $t_8 = 2.359$, $p = 0.0461$; Fig. 5b) and 24 h ($n = 6$/group, Student's $t$-test, $t_{10} = 2.503$, $p = 0.0313$; Fig. 5c) following the last self-administration session. There appeared to be specificity of the Tau phosphorylation since the Serine-199 residue that is not targeted by FYN was not significantly changed by heroin exposure ($n = 3/14$ control/heroin, Student's $t$-test, $t_{15} = 0.246$, $p = 0.8090$; Supplementary Fig. 11). Taken together, these data suggest that heroin-induced increases in FYN kinase lead to hyperphosphorylation of Tau. The consequent accumulation of phosphorylated Tau[29] might enhance vulnerability to cognitive impairments observed in long-term drug users.

Identifying FYN as the most significantly altered locus for chromatin impairments in striatal neurons in heroin abusers was also congruent with pathway analysis of the ATAC-seq peaks in which more open chromatin peaks from neurons also highlighted focal adhesion kinase (FAK)-associated intracellular signaling pathway as one of the most significantly altered (Supplementary Fig. 12). This finding is significant since FAK is a cytoplasmic tyrosine kinase that regulates cytoskeleton reorganization[30] and FYN is unique in the SRC family in binding specifically to FAK in mediating its effects relevant to cytoskeletal dynamics[31].

**Inhibition of FYN attenuates addiction-like behavior in rats.** Based on convergent evidence for heroin-related epigenetic and transcriptional alterations in relation to FYN obtained from our post-mortem human studies and the animal self-administration model, we hypothesized that FYN could be a promising non-opioid candidate for targeted therapeutic interventions in heroin users and thus could be examined in our preclinical model. Further support for this hypothesis comes from the fact that FYN inhibitors are already used and well tolerated in preclinical[26] and clinical studies of Alzheimer's disease[11].

In order to test *Fyn*'s role in addiction-like behaviors, adult male Long-Evans rats were trained to self-administer heroin. Upon reaching the maintenance phase, rats were intraperitonially injected with either the FYN inhibitor saracatinib (AZD0530, 5 mg/kg), or vehicle (2% DMSO, 30% PEG-300 in ddH$_2$O), 30 min prior to self-administration sessions. This dose, which was previously shown to rescue memory impairments in mouse models of Alzheimer's disease[26], was sufficient to significantly decrease Fyn kinase activity in the dorsal striatum of rats ($n = 8$/group, Student's $t$-test, $t_{14} = 3.697$, $p = 0.0024$ for active p-Fyn and $t_{14} = 4.055$, $p = 0.0012$ for inactive p-Fyn; Supplementary Fig. 13). Strikingly, the saracatinib treatment strongly and immediately attenuated heroin self-administration behavior on all three test days it was administered (two-way ANOVA, $n = 6$/group; $F_{1,29} = 6.852$, $p = 0.0139$, Fig. 6a). Inactive lever pressing (Fig. 6b) and general locomotor activity were not affected, and there were no group differences in active lever pressing prior to testing (Student's $t$-test, $t_{10} = 0.1205$, $p = 0.9065$) or 72 h after the last saracatinib injection (Student's $t$-test, $t_{10} = 0.0983$, $p = 0.9237$), all of which emphasize the specificity and acuteness of saracatinib's effect on heroin-taking behavior.

In order to determine saracatinib's effects on responding for heroin-associated stimuli (a behavior akin to drug-seeking in humans), animals were carried through an operant session with exposure to the previous heroin-paired cue light, but where lever pressing did not result in the delivery of heroin. Saracatinib significantly decreased responding for the heroin-paired cue (two-way ANOVA, $n = 6/7$ for vehicle/saracatinib, $F_{1,132} = 7.376$, $p = 0.0075$; Fig. 6c, d), which may be of translational relevance for human addiction since drug-seeking behavior strongly predicts relapse[32–34]. Interestingly, the saracatinib-induced decrease of active lever pressing during the three self-administration sessions when saracatinib was administered correlated with the decreased level of responding during the subsequent cue-induced test ($n = 13$, linear regression, $r = 0.67$, $p = 0.0124$; Supplementary Fig. 14C). This is likely not due to carry-over effects of behavioral adaptations as no group difference was observed during the last self-administration session (Fig. 6a). Nonetheless, future experiments where saracatinib is only introduced during abstinence are warranted to confirm that the drug has specific effects on cue-induced responding versus long-term depression of operant responding.

To determine generalized effects of saracatinib to natural reward, we studied the self-administration of palatable food reward. Saracatinib did not affect sucrose self-administration behavior (two-way ANOVA, $n = 11$/group; $F_{1,60} = 0.0344$, $p = 0.8535$; Supplementary Fig. 14A, B), emphasizing the specificity of the behavioral findings.

In the same cohort of animals, we observed that the described behavioral effects of saracatinib tended to be accompanied by decreased protein tyrosine kinase (PTK) activity ($n = 6$/group, Student's $t$-test, $t_{10} = 0.9368$, $p = 0.37$; Supplementary Fig. 15A) and Tau phosphorylation ($n = 6$/group, Student's $t$-test, $t_{10} = 1.798$, $p = 0.1024$; Supplementary Fig. 15B) in the caudate-putamen. These effects did not reach statistical significance, likely due to the non-specificity of the PTK assay used and the relatively short duration of treatment. However, dorsal striatal Tau phosphorylation correlated significantly with dorsal striatal PTK activity in these animals (linear regression, $F_{1,10} = 5.556$, $r = 0.598$; $p = 0.0401$; Supplementary Fig. 15C), suggesting that saracatinib-induced decreases in kinase activity did manifest in lower levels of phosphorylation of Fyn-specific targets, even after a relatively short duration (3 days) of treatment.

As saracatinib is a non-specific inhibitor of Src family kinases, we used an orthogonal approach to further assess the direct

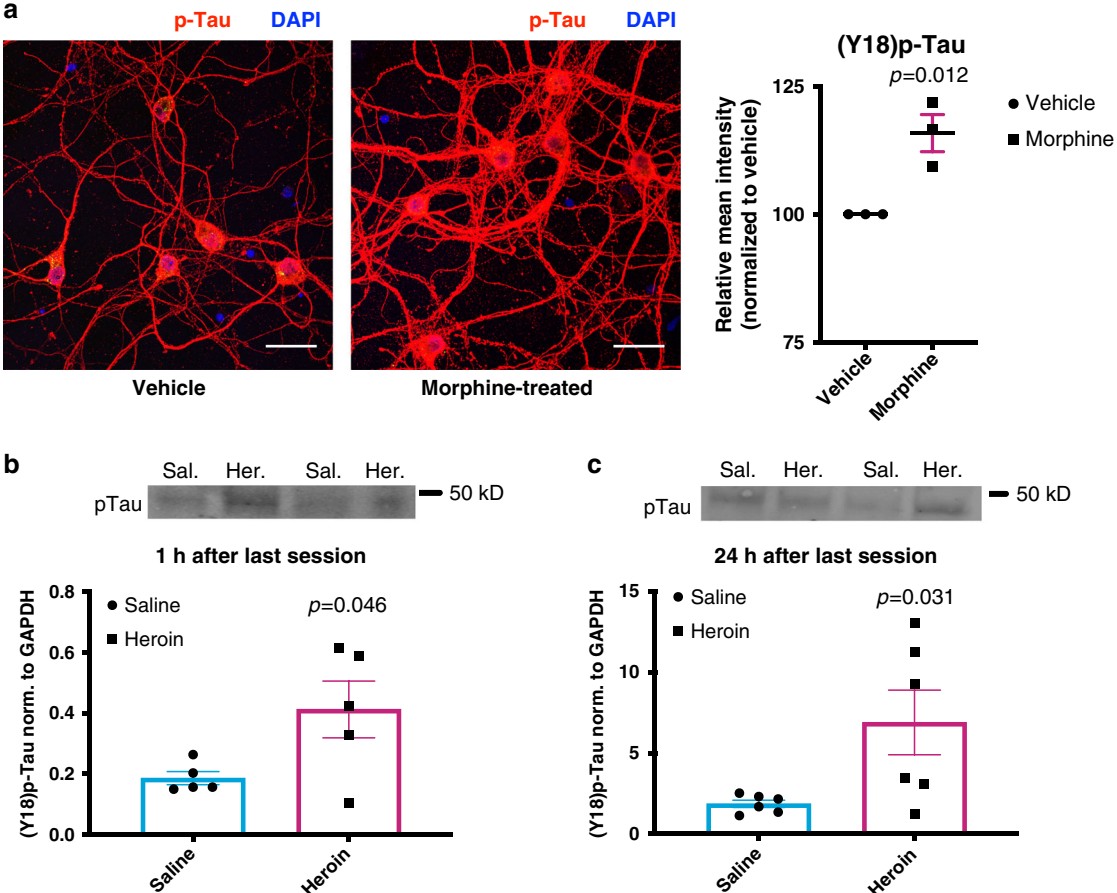

**Fig. 5 Heroin leads to FYN-mediated hyperphosphorylation of Tau protein. a** Immunocytochemistry analysis of vehicle and morphine-treated primary cortical-striatal co-cultured neurons revealed increased phosphorylation of Tyrosine-18 of Tau in morphine-treated cells ($n = 3$ biological replicates/group, Student's $t$-test, $t_4 = 4.374$, $p = 0.0119$). ×40 magnification, scale bars represent 33 μm. **b, c** Western blots from the caudate-putamen of heroin self-administering rats showed Tau hyperphosphorylation in vivo both 1 h (B; $n = 5$/group, Student's $t$-test, $t_8 = 2.359$, $p = 0.0461$) and 24 h (**c**; $n = 6$/group, Student's $t$-test, $t_{10} = 2.503$, $p = 0.0313$) following the last self-administration session. Data are represented as mean ± SEM. DAPI 4,6-diamidino-2-phenylindole. Source data are provided as a Source Data file.

contribution of Fyn. In rats that were previously trained to self-administer heroin, we performed siRNA-mediated knock-down of Fyn in the dorsal striatum (Supplementary Fig. 16), which resulted in significantly decreased mRNA levels of *Fyn* but not other Src family members 3 days following injection of 20 pmol siRNA ($n = 7$/group, Student's $t$-test, $t_6 = 3.822$, $p = 0.0087$; Supplementary Fig. 17). Importantly, this approach ensured specificity for Fyn and limited the manipulation to the dorsal striatum, the striatal area where heroin-induced molecular alterations were observed in the human and rat (Fig. 4). Strikingly, we found that in rats that were previously trained to self-administer heroin (Supplementary Fig. 18), dorsal striatal knock-down of Fyn strongly attenuated active lever responding induced by the light cue (two-way ANOVA, $n = 5$/group, $F_{1,96} = 4.007$, $p = 0.0481$; Fig. 6e, f) or drug-prime (two-way ANOVA, $n = 6$/group, $F_{1,120} = 8.545$, $p = 0.0041$; Fig. 6g, h). Of note, neither saracatinib nor siRNA-mediated knock-down of Fyn affected inactive lever pressing or general locomotor activity in these experiments (Supplementary Fig. 19).

Taken together, our data indicate that systemic pharmacological and dorsal striatal genetic inhibition of the FYN tyrosine kinase strongly and selectively decrease addiction-like behaviors and thus represent a promising option to explore for the development of targeted therapies of heroin-use disorder.

## Discussion

Using a genome-wide direct assessment of chromatin state in neuronal versus non-neuronal cells in the human brain, we identified the SRC family tyrosine kinase FYN as the most significantly affected locus in dorsal striatal neurons of heroin users. Focused transcriptional profiling of the *FYN* gene revealed increased mRNA expression in human heroin users, rats self-administering heroin, and primary neurons chronically exposed to morphine in vitro. In addition, heroin exposure increased dorsal striatal Fyn kinase activity in rats, as well as phosphorylation of the FYN target Tau, a process associated with neurodegeneration. Strikingly, inhibiting FYN kinase with saracatinib or siRNA potently decreased heroin reinforcement and cue-induced responding behaviors in a rodent self-administration model without impacting palatable food self-administration or general locomotor activity. Thus, our systematic interrogation starting from assessment of chromatin accessibility of the dorsal striatum in human heroin users directly links Fyn kinase activity to neurodegenerative-like Tau hyperphosphorylation in this region; and, crucially, demonstrates that pharmacological or genetic ablation of Fyn in this brain region attenuates heroin-taking and drug- and cue-induced responding, which is of translational relevance with respect to craving and relapse in human heroin users[34]. Although future studies are required to

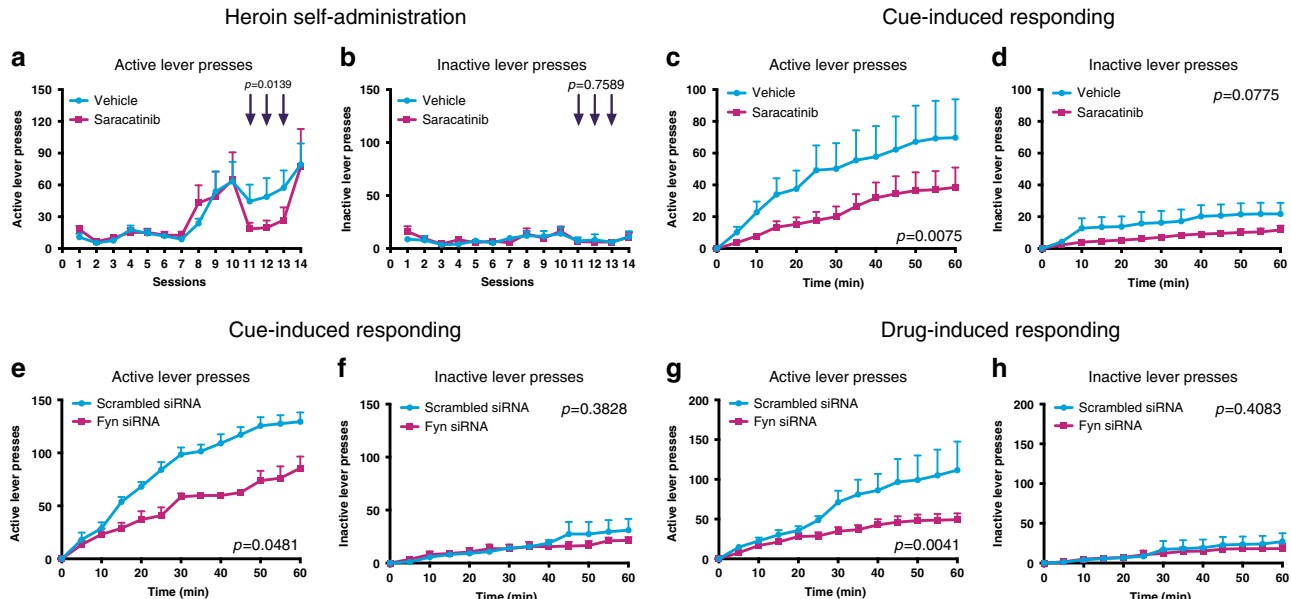

**Fig. 6 Inhibition of Fyn attenuates heroin self-administration and cue-induced responding. a** Active lever presses during 3 h long heroin self-administration session in vehicle and saracatinib treated Long-Evans rats. Arrows represent days of saracatinib administration. Two-way ANOVA revealed significant group effect ($n = 6$/group, $F_{1,29} = 6.852$, $p = 0.0139$) indicating decreased heroin self-administration behavior on days of saracatinib exposure. **b** Inactive lever presses were not affected in the same sessions ($n = 6$/group, two-way ANOVA, $F_{1,29} = 0.096$, $p = 0.7589$). **c** Active lever presses during a 1 h long, cue-induced, non-reinforced session in rats previously trained to self-administer heroin that were injected with vehicle or saracatinib ($n = 6/7$ for vehicle/saracatinib, two-way ANOVA, $F_{1,132} = 7.376$, $p = 0.0075$). Saracatinib significantly decreased responding for the heroin-paired stimulus. **d** Inactive lever presses were not affected in the same sessions ($n = 6/7$ for vehicle/saracatinib, two-way ANOVA, $F_{1,132} = 3.165$, $p = 0.0775$). **e** Active lever presses during a 1 h long, cue-induced, non-reinforced session in rats previously trained to self-administer heroin and subjected to siRNA-mediated knock-down (KD) of Fyn in the dorsal striatum 3 days prior to the session ($n = 5$/group, two-way ANOVA, $F_{1,96} = 4.007$, $p = 0.0481$). KD of Fyn significantly decreased responding to the heroin-paired cue. **f** Inactive lever presses were not affected in the same sessions ($n = 5$/group, two-way ANOVA, $F_{1,96} = 0.7685$, $p = 0.3828$). **g** Active lever presses during a 1 h long, drug-induced, non-reinforced session in rats previously trained to self-administer heroin and subjected to siRNA-mediated knock-down of Fyn in the dorsal striatum 3 days prior to the session ($n = 6$/group, two-way ANOVA, $F_{1,120} = 8.545$, $p = 0.0041$). KD of Fyn significantly decreased active lever pressing. **h** Inactive lever presses were not affected in the same sessions ($n = 6$/group, two-way ANOVA, $F_{1,120} = 0.6884$, $p = 0.4083$). Data are represented as mean ± SEM. Source data are provided as a Source Data file.

further delineate the role of Fyn inhibition on more nuanced aspects of these behaviors (e.g., habitual versus goal oriented responding for drug or cues), these novel findings indicate that FYN inhibitors may be promising therapeutic tools for heroin-use disorder.

FYN belongs to the SRC family of protein tyrosine kinases[35], is highly expressed in the human brain[36] and has an important role in the central nervous system[37]. Intriguingly, FYN localizes to the post-synaptic density of glutamatergic synapses and associates with NR2B, a subunit of the NMDA receptor[38], which is elevated in heroin abusers. This interaction is mediated by the scaffolding protein RACK1 (receptor of activated protein C kinase 1)[39,40] and leads to phosphorylation of NR2B[41–43]. FYN-induced phosphorylation of NR2B in turn enhances NMDA receptor function[38,40,43] by increasing surface expression in post-synaptic membranes[44–46], a cellular pathway that may mechanistically explain heroin's impact on striatal synaptic plasticity.

Glutamate receptor trafficking and function is well known to have a central role in neuronal adaptations that underlie the abuse of various substances[20,47]. Furthermore, long-lasting activation of Fyn has been previously described by Ron and colleagues during alcohol use[23–25]. This and our work focused on the anterior, medial part of the striatum and, while we show no differences in the nucleus accumbens, further studies will be necessary to address brain region specificity.

Intriguingly, saracatinib has been shown not only to inhibit alcohol-induced upregulation of FYN and NR2B, but to also suppress alcohol self-administration and seeking in mice[48].

Moreover, saracatinib decreases symptoms of physical withdrawal in mice implanted with morphine pellets[49], further emphasizing its relevance for opioid-related pathologies. Other SRC kinase inhibitors also attenuate instrumental responding for cocaine[50]. None of these inhibitors are, however, specific to FYN so the contribution of distinct SRC family members remains to be elucidated. Intriguingly, chromatin accessibility alterations in the post-mortem human brain were specific to FYN among members of the SRC family (Supplementary Fig. 10). Furthermore, siRNA-mediated knock-down of Fyn decreased heroin- or cue-induced responding (Fig. 6e–h). Thus, our data support an important role of Fyn in relation to heroin.

FYN is also strongly implicated in neurodegenerative diseases such as Alzheimer's disease. Specifically, binding of Aβ-oligomers to cellular prion protein triggers mGluR5-dependent signaling leading to the activation of FYN[27]. Importantly, Tau, the hyperphosphorylation of which has a central role in Alzheimer's neuropathology, is a direct substrate of FYN and can be phosphorylated by this kinase on its Tyrosine-18 residue[51], which was elevated with opioid exposure. Excessive hyperphosphorylation of Tau leads to synaptic dysfunction and loss[27], which strongly contribute to neurodegeneration and cognitive impairment.

We and others have previously observed hyperphosphorylated Tau in specific brain areas, including the striatum, of autopsy specimens from human heroin abusers[29,52–55]. Although pTau levels correlated with age in all subjects in certain predilection areas (e.g., entorhinal cortex, frontal cortex), this relationship was strongly exacerbated by heroin, suggesting accelerated

neurodegenerative-like pathology in heroin users that might contribute to drug-induced cognitive impairments. Our current in vivo and in vitro studies demonstrate a direct causal link between opioid exposure, enhanced activity of FYN and abnormal Tau phosphorylation. It is important to note that the heroin-related pTau immunohistochemical pathology we detected in heroin abusers[29] has a unique signature given that other neurodegenerative marks studied did not mirror those documented in various neurodegenerative disorders or tauopathies. For example, there was a lack of α-synuclein (e.g., Parkinson's disease) and TDP-43 (e.g., frontotemporal degeneration) immunoreactivities and no deposits of amyloid beta (e.g., Alzheimer's disease) were detected in the forebrain of heroin users[29]. Heroin users exhibit both transient and long-lasting cognitive impairments[56–58], but it is currently unknown whether this is accompanied by striatal or cortical neurodegeneration. Intriguingly, a recent study reported neurodegenerative-like mRNA expression profiles in blood samples from over 800 heroin-dependent subjects[59]. Whether observed pTau pathology represents an early sign of vulnerability to neurodegenerative disorders later in life or reflects the lower end of the Tau-related spectrum of cognitive impairment is unknown. Intriguingly, saracatinib attenuated neurodegenerative-like pathology in rats, although no complete reversal was observed due to the short duration of treatment. The mechanistic link we demonstrated between opioid exposure and dysregulated FYN and Tau pathology could also have marked implications for the large number of individuals exposed chronically to potent opioid analgesics during the past decade. Future experiments overexpressing Fyn will be important to further corroborate its direct role in Tau phosphorylation related to heroin-associated behavioral phenotypes.

The ATAC-seq of neurons and non-neuronal cells offered direct insights into the heroin-addicted human brain. Intriguingly, we revealed distinct perturbations of the chromatin landscape in the two cellular populations. This duality is consistent with our recent findings in the healthy human brain, indicating that chromatin state varies markedly by cell type[9]. Although the current study focused on neuronal impairments, non-neuronal populations were also strongly affected by heroin and potentially contribute to addiction pathology. For example, SLITRK1, the most significantly altered gene in the non-neuronal population, is closely linked to synaptic plasticity[60].

Overall, we report several lines of converging evidence with direct insights from the addicted human brain that was translatable to animal models of this disease. Such orthogonal approaches are imperative to improve current treatment strategies for neuropsychiatric disorders. Direct interrogation of the chromatin state of different cell types in the brains of heroin abusers provided evidence of discrete epigenetic dysregulation in the genome with greater impairment within striatal neurons emphasizing the potentially greater vulnerability of those cells to heroin. Identifying FYN as the most affected locus was surprising but provides a logical molecular mechanism to account for the unexpected hyper-pTau pathology we previously detected in heroin abusers and could presently link to the causal exposure to opioids in our in vivo and in vitro rodent models. Importantly, the fact that the FYN inhibitor saracatinib potently decreased addiction-like behaviors strongly suggests that FYN, and related network, is a key target for future therapeutic development. Although the potential efficacy of FYN inhibitors through current clinical trials for Alzheimer's disease remains to be determined, most drugs targeting FYN appear to be well tolerated and therefore could be excellent candidates to repurpose for accelerated clinical studies focused on substance use disorders. As other members of the SRC family of kinases do not have a similar role, more selective inhibitors of Fyn could also be developed based on the chemical structure of saracatinib, and used with perhaps even greater efficacy in heroin users. The development of such non-addictive and non-opioid therapies for opioid addiction is critically needed to help combat the current opioid epidemic.

## Methods

**Human brain samples (Supplementary Data 1)**. Human brains from a homogeneous cohort of Caucasian subjects from apparent heroin overdose and normal controls (determined by self-report and ancestral informative marker analysis)[61] were collected at autopsy within 24 h of time of death at the Department of Forensic and Insurance Medicine, Semmelweis University, Hungary[62]. Informed consent from legally authorized representatives was obtained at the time of autopsy. Cause of death was determined by the forensic pathologist performing the autopsy (Supplementary Data 1). All users in the heroin group died due to heroin overdose. Inclusion criteria for the heroin group were documented history of heroin use and/or positive toxicology at the time of death, physical evidence of i.v. drug use (needle tracks); while multi-drug users (as determined by history and blood/urine toxicology at autopsy), subjects receiving methadone/buprenorphine treatment, and subjects with head trauma and comorbid psychiatric diagnoses were excluded. For the control group, inclusion criteria were negative toxicology, no history or physical evidence of opiate or other drug use, no head trauma and no documented psychiatric disorders[2]. Univariate correlations for demographic variables (age, pH, gender, etc.) were explored and significant variables were included in the final model. Ventral striatal (nucleus accumbens) and dorsal striatal (putamen) tissue punches were obtained from the same coronal plane (Supplementary Fig. 15A). 10 heroin users and 10 control subjects were used in the ATAC-seq experiment; transcriptional profiling was performed using qPCR on 30 heroin users and 16 control subjects. Total FYN protein levels were assessed using western blot in 26 heroin users and 12 controls.

**Fluorescence-assisted nuclear sorting (FANS)**. We used an established protocol that combines FANS and ATAC-seq to generate open chromatin profiles in neuronal and non-neuronal nuclei[8]. Briefly, 50 mg of frozen post-mortem brain tissue was homogenized in cold lysis buffer (0.32 M Sucrose, 5 mM CaCl$_2$, 3 mM Mg (Ace)$_2$, 0.1 mM, EDTA, 10 mM Tris-HCl (pH = 8), 1 mM DTT, 0.1% Triton X-100) and filtered through a 40 μm cell strainer. The flow-through was underlaid with sucrose solution (1.8 M Sucrose, 3 mM Mg(Ace)$_2$, 1 mM DTT, 10 mM Tris-HCl, pH8) and subjected to ultracentrifugation at 24,000 rpm for 1 h at 4 °C. Pellets were thoroughly washed and resuspended in 500 μl DPBS, then incubated in BSA (final concentration 0.1%) and anti-NeuN antibody (1:1000, Alexa488 conjugated, Millipore MAB377X) under rotation for 1 h, at 4 °C, in the dark. Prior to FANS sorting, DAPI (ThermoScientific) was added to a final concentration of 1 μg/ml. Neuronal (NeuN+) and non-neuronal (NeuN−) nuclei, counter-stained with DAPI, were sorted using a FACSAria flow cytometer (BD Biosciences).

**ATAC-seq libraries**. ATAC-seq (assay for transposase accessible chromatin) reactions were performed using an established protocol[63] (Illumina Cat #FC-121-1030) and the resulting libraries amplified using the Nextera index kit (Illumina Cat #FC-121-1011). Optimal library amplification was determined by visualization with Bioanalyzer High Sensitivity DNA Chips (Agilent technologies Cat#5067-4626). Libraries were amplified for a total of 12–17 cycles and were quantified by Qubit HS DNA kit (Life technologies) and by quantitative PCR (KAPA Biosystems Cat#KK4873) prior to sequencing. Finally, ATAC-seq libraries were sequenced on Hi-Seq2500 (Illumina) obtaining 50 bp paired-end reads. All ATAC libraries met or exceeded ENCODE guidelines (ChrM < 1.5%, PBC > 0.5, RSC > 0.3, FRiP > 5%) (Supplementary Data 2).

**Real-time quantitative polymerase chain reaction (qPCR)**. Total RNA was extracted from approximately 10 mg of pulverized putamen tissue using the RNAqueous Micro Kit (Ambion) and following the manufacturer's protocol. First-strand cDNA was synthesized using qScript cDNA SuperMix (Quanta Biosciences, Gaithersburg, MD, USA) and subjected to qPCR analysis using TaqMan-based probes. Primers and probes against FYN (Hs00941600_m1 for human and Rn00562616_m1 for rat), GRIA1 (Hs00181348_m1), GRM5 (Hs00168275_m1), HOMER1 (Hs00188676_m1), and DLG4 (Hs00176354_m1) were obtained from Applied Biosystems (Life Technologies, Carlsbad, CA, USA). Eukaryotic 18S rRNA (Applied Biosystems, product # 4319413E) was included in each multiplex PCR as an internal control. Real-time PCR and subsequent analysis were performed with a Roche LightCycler 480 Real-Time PCR system (LightCycler 480 software; Roche, Basel, Switzerland). Quantification of target gene expression in all samples, performed in triplicate, was normalized to 18S rRNA by the equation $C_{T(target)} - C_{T\,(18S)} = \Delta C_T$, where $C_T$ is the threshold cycle number. Differences between control and heroin subjects, including individual variation, were calculated by the equation $\Delta C_{T\,(individual\ subject)} - \Delta C_{T\,(mean\ control)} = \Delta\Delta C_T$. Changes in target gene expression (n-fold) in each sample were calculated by $2^{-(\Delta\Delta CT)}$ from which the means and standard errors of the mean (SEM) were derived.

**Primary striatal culture and cortical-striatal co-culture**. Primary rat striatal cultures and striatal-cortical co-cultures were generated from Long-Evans embryos[64,65]. In brief, pregnant Long-Evans dams were euthanized with continuous $CO_2$ and embryos were carefully removed by Cesarean section at embryonic day 18.5 (E18.5). Striatal and cortical tissues were dissected, separately trypsinized for 15 min at 37 °C, washed three times with room temperature HBSS for 5 min, and passed through a 40 μm cell strainer to remove debris. Primary cells were plated 3:2 (cortical:striatal) or striatal only at a density of $2 \times 10^5$ cells per glass coverslip, which were coated with poly-L-lysine-coated and embedded on a 35 $mm^2$ dish. Cells were plated in plating media (Minimum Essential Media containing 1 mM sodium pyruvate, 20% glucose, 10% fetal bovine serum, N-2 supplement, and Penicillin–Streptomycin). On the next day, plating media was replaced with maintenance neurobasal media (1 mM sodium pyruvate, B-27 and N-2 supplements, and Penicillin–Streptomycin). Cells were treated with a mitotic inhibitor (Ara-C, 5 μM) on the fourth day in vitro (DIV4) and half the maintenance media was exchanged with fresh media every 3–4 days until completion of the experiment. Cells were treated with 10 or 100 μM morphine every other day for one week (from DIV8 to DIV14) and fixed 24 h after the last dose with 4% PFA in DPBS (with 4% sucrose) for 10 min at room temperature. These levels of morphine are physiological and routinely used in the literature. Importantly, concentrations in this range are sufficient to induce alterations in synaptic plasticity consistent with known effects observed in human heroin abusers and reported in animal models of heroin exposure[66–68]. Primary striatal:cortical co-cultures were used for immunocytochemistry; primary striatal cultures were used for real-time quantitative polymerase chain reaction.

**Immunocytochemistry**. After fixation, cells were washed with DPBS, permeabilized in 0.25% Triton X-100 in PBS for 5 min, and blocked in 10% BSA in DPBS for 1 h. Cells were incubated with primary antibodies recognizing pTau-Y18 (mouse mAb at 1:200, cat no. GTX54658 from GeneTex, CA, USA) and DARPP-32 (rabbit mAb at 1:100, cat no. 2306S from Cell Signaling Technology, Danvers, MA) diluted in DPBS with 3% BSA. Cells were washed three times with DPBS and AlexaFluor-488 secondary antibodies (Thermo Fisher A32731 and A32723) were diluted 1:1000 in DPBS with 3% BSA for subsequent 1-h incubation at room temperature. Finally, cells were washed three times with DPBS and briefly rinsed with water before mounting in hard-set media containing DAPI (Vector Laboratories, Burlingame, CA). Striatal medium spiny neurons were selected on the basis of DARPP-32-immunoreactivity then imaged on a Zeiss LSM780 inverted confocal microscope at ×40 magnification. Quantification of images was performed by a researcher blind to the treatment condition using Fiji (2017) for processing and MATLAB9.7 to generate the average intensity.

**Western blot**. Approximately 15 mg of pulverized dorsal striatum (putamen) punches per subject were used to generate histone extracts using 100 μl 0.2 N HCl containing protease (cOmplete Mini, EDTA-free protease inhibitor cocktail, Roche) and phosphatase inhibitors (Halt phosphatase inhibitor cocktail, ThermoScientific, Rockford, IL, USA) and 5 mmol/l Na-butyrate. The protein was diluted in Laemmli buffer, denatured (for 5 min at 95 °C), subjected to electrophoresis, and transferred onto nitrocellulose membranes. The membranes were blocked in Odyssey blocking buffer (LI-COR, Lincoln, NE, USA) and incubated at 4 °C overnight with primary antibodies. Rabbit polyclonal antibody was used against p(Ser199)-Tau (44734G, Thermo Fisher, dilution 1:1000) and Fyn (4023S, Cell Signaling, dilution 1:1000) and mouse monoclonal antibody was used against p (Y18)-Tau (GTX54658, GeneTex, dilution 1:1000). Membranes were subsequently incubated with goat anti-rabbit or goat anti-mouse IRDye 680 (LICOR, 926-68071), or IRDye 800 (LICOR, 926-32210) secondary antibodies diluted 1:5000 at room temperature for 1 h. Membranes were developed with the LICOR infrared imaging system (LICOR) and images quantified using average integrated density values. GAPDH (MAB374, Millipore, diluted 1:5000) levels were used to control for total protein content. All uncropped western blots are shown in Supplementary Figs. 20 and 21.

**Kinase activity measurements**. Rat dorsal striatum tissue was homogenized using BioRuptor in 1× IP buffer (150 mM NaCl, 10 mM Tris-HCl pH = 7.4, 1 mM EDTA, 1 mM EGTA, 1% Triton X-100) containing protease and phosphatase inhibitors. Homogenates were pre-cleared using protein G Dynabeads for 1 h at 4 °C. For immunoprecipitation, 1 mg of pre-cleared protein was incubated overnight at 4 °C with 7.5 μg of total Fyn (4023S, Cell Signaling, diluted 1:1000) or 5 μg of total Src (ab16885, Abcam, diluted 1:1000) antibody. Immunocomplexes were pulled down with protein G Dynabeads for 4 h at 4 °C and eluted into Laemmli buffer at 55 °C for 10 min. 50 μg of immunoprecipitated protein was used for western blotting. To determine kinase activity, antibodies against active p(Y418)-Src (ab4816, Abcam, diluted 1:200) and inactive p(Y529)-Src (ab32078, Abcam, diluted 1:1000) were used. In addition, dorsal striatal protein tyrosine kinase activity was also measured using a commercially available kit following the manufacturer's instructions (Omnia Kinase Assay, KNZ3051, Thermo Fisher).

**Human FYN luciferase plasmid subcloning**. Primers for amplifying the putative human *FYN* regulatory region (hg19_dna range=chr6:112232247-112232896)

were designed using Primer3 version 0.4.0. Restriction enzymes sites for *Kpn*I and *Mlu*I were chosen from the multiple cloning site on Promega's pGL3 promoter vector (cat no. E1761). Forward and reverse primers with *Kpn*I (GGTACC) and MluI (TGCGCA) restriction sequences, and leader sequences (TAAGCA/ ATTCGT) were as follows: Forward 5′-TAAGCAGGTACCCATGTAAAAAGT GATTATCAGAAGAGG-3′. Reverse: 5′-TGCTTAACGCGTCTTGTTGCTCTAC TCTCCTCAGC-3′. The human putative Fyn regulatory region was amplified using human cerebellar genomic DNA. The PCR conditions were as follows for 20 μl reactions: 250 nM pooled primers, 1 μl of human gDNA (~100 ng), 0.5 μl MilliQ water, and 18 μl of Invitrogen's Platinum PCR Supermix High Fidelity. PCR cycles were 94 C for 2 min, 39× (94 °C for 15 s, 62.2 °C for 30 s, 68 °C for 30 s), 4 °C for infinite hold. The insert sequence was verified by Sanger sequencing. 1 μg of insert and pGL3 vector were double digested with *Kpn*I and *Mlu*I for 15 min at 37 °C, followed by purification with Qiagen's PCR Purification kit. The human Fyn insert was ligated into the pGL3 promoter vector with Invitrogen's Stabl3 *E. coli* following manufacturer's instructions under 3:1 insert to vector ratio. The bacterial cells were plated on 100 μg/ml ampicillin Agar plates overnight for 16 h at 37 °C. The next day, single colonies were selected with a pipette tip and inoculated into 5 ml LB broth and 100 μg/ml ampicillin and shook overnight at 250 rpm for 16 h. Plasmid DNA was isolated using Qiagen's Miniprep kit. The isolated plasmid DNA was then verified by Sanger Sequencing for presence of the human Fyn insert. Maxipreps were prepared by combining 250 ml LB Broth with 100 μg/ml ampicillin and 500 μl of starter culture, followed by shaking at 250 rpm at 37 °C for 16 h. Plasmid DNA was isolated using Qiagen's Endotoxin-free Maxiprep kit and the hFyn-pGL3 vector was verified by Sanger sequencing.

**HEK293T cell maintenance and transfection**. HEK293T cells were passaged into 75 $cm^2$ flasks and maintained in high glucose DMEM with 10% FBS and without Penicillin–Streptomycin. The day before transfection, cells were trypsinized and seeded into 24-well plates at 25,000 cells per well. On the day of transfection, cells were co-transfected with pGL3 control vector and pRL-SV40 vector (Promega, E2231) or hFyn-pGL3 vector and pRL-SV40 vector at a 10:1 ratio of pRL-SV40 vector to pGL3 vector in 200 ng total DNA per well. 1.5 μl of Lipofectamine 2000 reagent was used per well. Cells were then placed in a 37 C with 5% $CO_2$ incubator for 48 h.

**Dual-Glo luciferase assay**. Promega Dual-Glo Luciferase Assay reagents were prepared as per manufacturer's instructions (cat. no. E2920). 48 h post transfection, growth media was removed. 200 μl of crude cell lysis buffer (0.1% Triton X-100, 20 mM MgCl in 1× PBS) was added into each well. After mechanical titration with a pipette, the plate incubated on a shaker at room temperature for 10 min. 50 μl of cell lysate were transferred into a 96 well plate, followed by 50 μl of Dual-Glo Luciferase Reagent. Plate incubated at room temperature for 10 min on a shaker, followed by Firefly luciferase luminescence reading on a luminometer. 50 μl of Dual-Glo Stop & Glo Reagent was then added to each well. Plate incubated at room temperature for 10 min on a shaker, followed by Renilla luciferase luminescence reading on a luminometer.

**Rodent behavior and injections**. Adult male Long-Evans rats were maintained on reversed 12 h dark/light cycle and underwent jugular vein catheterization. They were subsequently food restricted (18 g/day during the initial training phase) and trained to self-administer heroin via the jugular catheter to control their own drug intake. Animals were placed in an operant chamber (29.5 cm × 32.5 cm × 23.5 cm) housed in sound-attenuating boxes (MED Associates Inc., St. Albans, VT) with two levers; depression of one (designated the active lever) resulted in the delivery of 30 μg/kg heroin under a fixed-ratio 1 (FR1) schedule of reinforcement, whereas depression of the other (designated the inactive lever) had no programmed consequences. Each self-administration session was 3 h long and performed in the morning (8 am to 11 am). Group sizes for each cohort are reported throughout the text. Each cohort underwent 7 FR1 sessions, followed by 4 additional FR5 sessions. The only exceptions were the animals shown in Fig. 4f–j, which underwent 11 FR1 sessions. Rats were food restricted for the initial 3–5 days and were fed ad libitum for the remainder of the study. Animals were sacrificed 1 h, 24 h or 2 weeks following the last self-administration session. For the saracatinib (AZD0530, AstraZeneca, Cambridge, UK) experiments, only rats that consistently differentiated between active and inactive levers (at least 10 active lever presses, at least 2× active as inactive lever presses on three consecutive days) and showed ~5x increased active lever presses under FR5 schedule compared to FR1 were selected for study. Animals received 5 mg/kg saracatinib i.p. injections 30 min before heroin self-administration testing on three consecutive days. To test for potential protracted effects of saracatinib, animals underwent an additional heroin self-administration sessions 24 h after the last saracatinib delivery. To test saracatinib's effect on responding for drug-paired cues, the last heroin self-administration was followed by 1 week of forced abstinence, then animals were injected with saracatinib and underwent a 1 h non-reinforced, cue-induced session. For acute heroin exposure, naive rats were injected with 1 mg/kg of the drug and sacrificed 1 h later. For acute saracatinib exposure, naive rats were injected with 5 mg/kg of the drug and sacrificed 30 min later. Protocols were approved by Icahn School of Medicine's review board. All testing was carried out using a counter-balanced

experimental design and conducted at a similar time of the day. Self-administration of a palatable food reward (unflavored sucrose pellets) was performed similarly using the same operant chambers in adult, male Long-Evans rats maintained on 12 h dark/light cycle. Rats were food restricted for the initial 3–5 days and were fed ad libitum for the remainder of the study. Food self-administration sessions were 1 h long and performed in the morning (8 am to 9 am). Each animal underwent 9 FR1 and 7 FR5 sessions.

**siRNA-mediated knock-down of Fyn**. In rats previously trained to self-administer heroin, guide cannulae were implanted at a 10° angle at AP + 1.7 mm, ML + 3.5 mm and DV + 2 mm from bregma (Supplementary Fig. 16B). Mouse Fyn siRNA (sc-35425) or control scrambled siRNA-A (sc-37007) from Santa Cruz were diluted to 20 μM working solutions in DNase/RNase-free water. Fyn siRNA (20 pmol) or control scrambled siRNA (20 pmol) were prepared with Polypus jetSI in vivo transfection reagent (cat no. 403-05) as per manufacturer's instructions. The final siRNA and jetSI mixtures were incubated at room temperature for 30 min. 1 μl of Fyn siRNA or control siRNA was infused into the dorsal striatum over 1 min, with two injection sites per hemisphere (28 gauge needles were lowered through the guide cannulae to AP + 1.7 mm, ML + 3.5 mm and DV + 5.2 mm from bregma, and subsequently pulled back to AP + 1.7 mm, ML + 3.5 mm and DV + 4.2 mm from bregma for another injection; Supplementary Fig. 16C). New siRNA mixtures were prepared after the completion of each infusion, per animal. 3 days post fusion, rats were subjected to drug- (0.25 mg/kg heroin i.p.) or cue-induced responding. Brains from sacrificed rats were flash frozen in isopentane and stored in −80 °C. Brains were dissected on a cold block set at −20 °C. Using a brain sectioning mold, frozen brains were sliced into 1 mm sections and the dorsal striatal tissue immediately at the visible injection sites were micropunched out. Isolated tissue was later subjected to total RNA extraction by TRIzol and iso-propanol precipitation. cDNA libraries were generated using the Quanta Bios-ciences XLT cDNA Synthesis kit. Rat Fyn and mouse Fyn mRNA expression were measured using Taqman primers (Rn00562616_m1). Other Src family genes were also quantified: Src (Rn00676848_m1), Yes (Rn01418794_m1).

**Statistical analysis**. Statistical evaluation was carried out using JMP10 (SAS Institute, Cary, NC, USA) or Prism8 (GraphPad Software, San Diego, CA). Data were tested for normality and normalized by natural log transformation, if not normally distributed. Univariate statistical analyses were used to study the effect of each independent demographic variable (e.g., age, gender, post-mortem interval, brain pH, RIN, years of heroin use, former overdose) and toxicology data (e.g., blood and urine alcohol, morphine, 6-mono-acetyl-morphine, codeine levels) on mRNA expression or protein levels. Variables with a $p$-value of <0.1 were included in the final multiple regression model. If there were no such variables, group differences were determined by one-way analysis of variance. Correlational ana-lyses were carried out according to Pearson.

**ATAC-seq data processing**. Briefly, reads were mapped to the gender appropriate reference genome hg19 using STAR aligner version 2.5.0, by disabling spliced alignments (setting --alignIntronMax 1) and prohibiting soft-clipping of the reads (setting --alignEndsType EndToEnd). Reads that (1) mapped to more than one locus, (2) were duplicated, and/or (3) mapped to the mitochondrial genome were excluded. Model-based Analysis of ChIP-seq (MACS) v2.1 was used to call peaks without building the shifting model (setting --nomodel). We retained peaks passing a $p$-value cutoff $4.7 \times 10^{-6}$ based on Irreproducibility Discovery Rate (IDR) ana-lysis[69] on technical replicates. Overall QC included: more than 10 million unique, non-redundant reads; <5% of reads mapping to the mitochondrial genome; more than 0.5 for PCR bottleneck coefficient, which is an approximate measure of library complexity estimated as (non-redundant, uniquely mapped reads)/(uniquely mapped reads); more than 0.3 for relative strand cross-correlation coefficient, which is a metric that uses cross-correlation of stranded read density profiles to measure enrichment independently of peak calling; more than 5% of reads in peaks, which is the fraction of reads that fall in detected peaks; more than 10,000 detected peaks; no outliers or admixture of cellular consistency based on principal component analysis (PCA). Overall, we excluded four samples due to low number of called peaks (low signal to noise ratio), cellular admixture and outlier based on PCA. Open chromatin regions were defined in each sample using the MACS2 algorithm as described above. Artifact regions (defined as "Blacklist" regions by ENCODE) with excessive unstructured anomalous reads mapping including regions at centromeres, telomeres and satellite repeats were excluded. Peak sets across cell types (neuronal and non-neuronal) and disease group (heroin and control) were consolidated into four single lists by union operations of peaks that were present in at least two libraries. The final consensus set of 143,003 peaks was generated by combining the four sets of peaks (neuronal control; non-neuronal control; neuronal heroin; non-neuronal heroin).

We used the featureCounts function from the Rsubread package[70] to generate a sample-by-peak read count matrix (36 samples by 143,003 peak regions). We counted fragments (defined from pair-end reads), instead of individual reads, that overlapped with the final consensus set of peaks. Scaling normalization to remove composition biases in sequencing data was applied to log(CPM) (Counts Per Million total reads) using the trimmed mean of $M$-values method[71].

We performed PCA of the normalized read count matrix and examined which variables were significantly correlated with the high-variance components (explaining > 1% of the variance) of the data. This identified four covariates, which encompassed seven statistical degrees of freedom (df): Cell type (neuronal and non-neuronal), Gender (male and female), Disease (control and heroin) and post-mortem interval (PMI). Gender and PMI clearly affect chromatin accessibility and are included as covariates in the differential analysis that examined disease and cell-type effects.

For each open chromatin region, the variance was decomposed into variation attributable to Cell type, Disease Group, Subject, Gender, PMI, plus the residual variation based on linear mixed models implemented in variancePartition v1.4.1 package[72]. The input to variance partition is counts per million ($\log_2$ CPM) matrix. The categorical and continuous variables were modeled as random and fixed effects, respectively. We initially fitted a model considering all variables (model 1). Because the majority of variance is attributed to cell type, we then modeled within/between Disease variance separately for each Cell type (model 2). We finally model the variance of Disease Group, Subject, Gender and PMI in separate models that considered only neuronal (model 3) and non-neuronal (model 4) samples. Each peak was considered separately and the results for all peaks were aggregated afterwards. Results were summarized in terms of the fraction of total variation explained by each variable for each peak.

We used the edgeR package[71] to model the normalized read counts using negative binomial distributions. The estimateDisp function was used to estimate an abundance-dependent trend[73]. To normalize for compositional biases, the effective library size for each sample was estimated using the trimmed mean of $M$-values approach as described above. We performed separate analyses for neuronal and non-neuronal samples. For each open chromatin region, we applied the following model for the effect on chromatin accessibility of each variable on the right-hand side: chromatin accessibility ~ Disease + Gender + PMI. Then, for each open chromatin region, the disease coefficient (heroin and control) was statistically tested for being non-vanishing, implying an estimated effect for disease, above and beyond any other effect from the covariates (Gender and PMI). A quasi-likelihood $F$-test was conducted using the glmQLFTest function[74] from the edgeR package, with robust estimation of the prior degrees of freedom. $p$-values were then adjusted for multiple hypothesis testing using false discovery rate estimation.

Enrichment analysis of peaks with genomic regions was done using R package LOLA (Locus Overlap Analysis) with Fisher's exact test[75]. $p$-values of enrichment analysis were then adjusted by Bioconductor qvalue package[76]. Genomic regions were collected from the following data sources. Transcription factor binding sites (TFBS) were downloaded from ENCODE[77], CODEX[78], and Cistrome[79]. Enhancer segments and repressed segments were downloaded from ENCODE. Tissue clustered DNase hypersensitive sites were downloaded from Sheffield et al[80]. CpG islands, microsatellites, genome repeat regions etc. were obtained from UCSC Genome Browser database. Transcription factor motifs were obtained from JASPAR[81]. Epigenomics modification regions of various conditions were downloaded from Cistrome and ROADMAP[82]. Genome enrichment/depletion analysis was performed using Homer[13].

**Reporting summary**. Further information on experimental design is available in the Nature Research Reporting Summary linked to this paper.

## Data availability

The sequencing data are available in GEO under accession code GSE136042. All other relevant data supporting the key findings of this study are available within the article and its Supplementary Information files or from the corresponding author upon reasonable request. Uncropped and unprocessed gels are included in Supplementary Figs. 20 and 21. Genomic regions were collected from the following data sources. Transcription factor binding sites (TFBS) were downloaded from ENCODE[77], CODEX[78], and Cistrome[79]. Enhancer segments and repressed segments were downloaded from ENCODE. Tissue clustered DNase hypersensitive sites were downloaded from Sheffield et al.[80] [http://dnase.genome.duke.edu]. CpG islands, microsatellites, genome repeat regions etc. were obtained from UCSC Genome Browser database. Transcriptional factor motifs were obtained from JASPAR[81] [http://jaspar.genereg.net]. Epigenomics modification regions of various conditions were downloaded from Cistrome. A reporting summary for this Article is available as a Supplementary Information file. Source data are provided with this paper.

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

## Acknowledgements

This work was supported by the National Institutes of Health—R56DA015446 (Y.L.H.), R01DA15446 (Y.L.H.), P01DA047233 (Y.L.H.), F30DA038954 (M.L.M.), R01AG050986 (P.R.), R01MH109677 (P.R.), Brain Behavior Research Foundation (20540 P.R.), and the Veterans Affairs (Merit grant BX002395 P.R.). G.E. is supported by the Alzheimer's Association (AARF-19-618159). Further, this work was supported in part through the computational resources and staff expertise provided by Scientific Computing at the Icahn School of Medicine at Mount Sinai.

## Author contributions

G.E., Y.L.H., and P.R. conceived the project. G.E. performed most of the experiments. J.F.F. performed FANS and ATAC-seq library preparation. G.E., J.E.C., J.-M.N.F., A.L., and J.A.L. performed heroin self-administration. D.A., T.R., and N.W. performed immunocytochemistry. D.A. performed IP-Westerns, TR performed cloning and luciferase assay. M.E.H., M.L.M., X.Z., B.Z., G.H., R.E., and P.R. analyzed the data. E.K. provided the brain samples. G.E. and Y.L.H. wrote the manuscript, which was read and approved by all co-authors.

## Competing interests

The authors declare no competing interests.
