## [Peer Review File · Nature Communications]

Reviewers' comments:

Reviewer #1 (Remarks to the Author):

This is an elegant and well conducted study that ventured into uncharted territories. The authors should be commended for the breadth of the methods deployed in this study and the originality of the investigations on human brains.

I have to acknowledge that I have limited expertise in epigenetics-related methods so my comments will be more related to other aspects of the study which suffers from several caveats and pitfalls that warrant a thorough revision prior to being considered further for publication.

1. What is the evidence for neurodegeneration-induced protracted cognitive or emotional deficits in heroin addicts? Is it not the case that cognitive deficits substantially decrease following abstinence in former heroin addicts? And what is the evidence that neurodegeneration occurs in the striatum of heroin users or heroin addicts (where the alterations of FYN activity were found in this study)?

2. Saracatinib is not selective for FYN. FYN is also involved in the trafficking of glutamate receptor subunits and has been shown to be potentially involved in alcohol addiction in humans as well as the reinforcing and incentive properties of cocaine in rats. Can the authors better discuss their findings and their originality in light of the literature on FYN and addiction?

3. It is a pity that the legends of fig 1 & 2 (and to some extent those of all the other figures) overall are a compilation of titles for each panel but carry no information or offer no take-home message. This will make the reading of the paper challenging to the broad readership of Nature Communications.

4. In the introduction the authors equate heroin use as a disease (line 1-2). Is this really what they mean?

5. The dorsal striatum is not involved in reward, is it? Where exactly in the dorsal striatum did the authors measure chromatin accessibility? Clearly the pDMS and the aDLS have opposite functional roles in drug seeking. This should be taken into account and the schematic representation of the brain samples would be much more useful if added to one of the key figures of the paper.

6. Are the epigenetic differences observed between heroin users and controls associated with addiction or instead predate heroin use in the former, and hence represent a factor of vulnerability to, instead of the suggested biological basis of, addiction? Or merely a response to chronic exposure to heroin that is not related to addiction?

7. Were the heroin users exclusively using heroin? Did they not also use alcohol or cocaine (which have both been shown to be associated with alterations of Fyn activity?)

8. The authors use interchangeably drug seeking, drug self-administration and reinstatement of an extinguished response for drugs. Clearly each of these processes is mediated by different psychological mechanisms and neither self-administration (which measures reinforcement or taking, as the instrumental response is conflated with the consummatory response under fixed ratio schedules of reinforcement) nor reinstatement (which is the acquisition of responding for the conditioned reinforcing properties of drug-paired cues under extinction, hence in the absence of the drug, over a single session following extinction) measure drug seeking behaviour. Some clarification is needed in the manuscript as some authors go to great lengths to actually measure drug seeking behaviour or its compulsive nature in preclinical models. I suggest the authors dampen the claims re translational value of these procedures (which have none as far as I understand the psychological mechanisms that subserves the performance of rats in these tasks) or any reference to craving: if lever pressing under extinction is a measure of craving then not giving

the lever, hence preventing the animal from responding, is the best way to abolish craving! We should be careful not to overstate what behavioural procedures measure in species that are not endowed with language.

Overall the preclinical data are relatively weak: they show that manipulations of Fyn influence the reinforcing properties of heroin.

9. Is the decreased level of responding observed at the abstinence test predicted by the overall instrumental output over the last 5/6 sessions of training (influenced by the same manipulations of FYN activity)

10. For the acquisition of heroin SA under FR schedule, what is the escalation the authors refer to? Do they mean that the rats displayed a progressive increase in infusions obtained per session which reveals that progressive acquisition of heroin reinforcement at the population level? What is the evidence there is escalation of intake?

11. For CS-induced reinstatement, (fig 6 A-F), since performance under reinstatement conditions occurs under extinction, the actual influence of the contingent presentation of the CS on responding can only be measured over the first 10 to 20 min of the reinstatement session in comparison with performance over the first 10 to 20 min of the previous extinction session (within-subject design with session as 1 within-subject factor). But the authors did not perform a reinstatement test as the rats had not been exposed to extinction beforehand. Instead they carried out a relapse challenge under extinction. Here the cumulative responses suggest there may be a difference between both groups in their ability to extinguish the influence of the CS over responding (this seems to be a real issue with the siRNA where the scrambled and target siRNA groups show little if no differences in the first 10 min of the reinstatement test). It would be more appropriate to present the actual active and inactive lever presses and carry out an analysis that factors the lever factor in as it seems the drug at the dose used (but clearly not the siRNA) also decreases IL presses (is it a general behavioural suppression?)

12. Also, can the authors better justify the use of the dose of saracatinib they used in this study?

13. How do the authors account for the direct effect of morphine on neuronal FYN in cultures? Do the authors suggest these effects are independent of dopamine in the intact striatum? This should be better discussed.

14. Some titles of the supplementary figures suffer from stylistic issues, for instance one refers to the dose of the drug, but not the drug itself...

Reviewer #2 (Remarks to the Author):

This study identifies differential accessible chromatin regions via unbiased ATAC-seq in human post-mortem brain tissue. This analysis revealed a putative regulatory region of the FYN locus, a gene previously implicated in neurodegenerative disorders, as more accessible in a patient population of heroin abusers. Activity of FYN and its downstream target Tau were found to be increased in post-mortem human striatum, rats self-administering heroin, and cultured striatal neurons treated with morphine. Manipulation of FYN via pharmacological and genetic strategies altered self-administration and reinstatement behaviors in a rodent model. Although the variety findings in human post-mortem tissue and model systems very interesting, there are a number of weaknesses described below that need to be addressed before this manuscript can provide a novel, particularly translational finding to the field.

Concerns:

1. The ATAC-seq data does not appear to be rigorously examined and is overall poorly laid out.

a. The QC standards provided in the methods does not indicate if there were any differences between heroin and control samples. The comments below might indicate that there is a technical issue between groups. It is also not indicated in the figure caption how many patient samples derive each sample.

b. It is not clear how the adjusted p-values were calculated and if it was corrected for genome-wide multiple comparisons.

c. Figure 1a is poorly labeled and should clearly indicate data tracks for both control and heroin samples. There are no genomic coordinates or scale bars either. The text is very small and is very hard to read.

d. Figure 1b is confusing in that what appears to be the same p-values are represented in 2 different colors. It is also noteworthy that the peaks in the heroin glia also follow the same trend as the neuronal peaks from heroin users in that they too are enriched in CpG islands, promoters, and 5'UTRs. The authors should comment about this and indicate if there is a peak detection issue in the control samples within the dataset.

e. Figure 1c is not very clear based on the figure caption. Do the numbers within the heat map correspond to the number of overlapping peaks? Additionally, the publicly available enhancer datasets from ENCODE are from cell lines and are likely to be different between cell culture lines and neurons given cell-type chromatin landscapes, so it is unclear if the overlap is of biological importance. Additionally, if the numbers within the heat maps are the number of peaks overlapping with the specific chromatin mark, then it is alarming that that there would be so few overlapping peaks corresponding to active chromatin in the neuron control and that heroin use alone would "suggest that transcriptional activity might be overall increased in neurons of heroin users." It seems that there are overall more peaks detected in the neuron heroin group from the observation that there are many more overlapping peaks detected across all tested regulatory elements in Figure 1c.

f. The text in Figure 2 is overall small and hard to read. Most panels are not labeled in a way that is intuitive to understand.

g. It is confusing that in Figure 1, the authors conclude that the open chromatin regions in neurons from heroin user implies that there is increased transcriptional activity in those cells. Figure 2 then describes that the less accessible chromatin regions in neurons involve transcription factor activity and RNA polymerase II activity. There is no matching GO-term analysis for the non-neuronal cells even though it is mentioned that non-neuronal regions largely overlapped with neuronal regions.

h. It is unclear whether the variance explained in Figure 2 significant between the different conditions (cell type, disease, gender, etc.). This should be clarified to understand if it is biologically meaningful or random variation.

2. The variance peak observed ~38kb upstream of the FYN locus is not adequately validated as an enhancer of that gene. It should be functionally validated as proximity does not imply that it is an enhancer of the FYN gene.

a. The raw ATAC signal shown in Figure 3a largely looks similar between control and heroin samples, but the marginal increase appears to be mostly in the control group over the heroin group.

b. The luciferase assay indicates that the region can drive increased gene expression. This could be the result of having an expanded promoter in the luciferase plasmid as there was not a control sequence (scrambled or otherwise) inserted as a control to correct for increased promoter length.

c. Additionally, increased luciferase activity does not indicate that the region is a definitive enhancer of the FYN locus. To establish this locus as an enhancer, it should be functionally tested with a deletion or inactivation, even if only in a cell line.

3. Figures 4 and 5 present interesting observations regarding the levels of FYN mRNA and protein in a variety of model systems.

- a. Figure 4a displays the promoter of FYN over the putative enhancer in Figure 3. There appears to be an increase in raw signal between control and heroin neuron samples, but this is not indicated as significant and could be the case if more peaks were detected in the neuron heroin group as mentioned in a previous concern.
- b. The data in Figure 4f-q appears to be normalized to the saline group, which should be clarified. It is not indicated if total levels of FYN protein are altered based on the way the data is normalized. If there is increased transcription and enhancer activity, it would be important to measure and display the levels of total protein. Also, the authors should comment on how potentially increasing the protein itself also results in more activated protein.
- c. Though it may not take away from the experimental interpretation, there were multiple schedules of self-administration sessions described in the methods for Figure 4. It is unclear why.
- d. Replicates are not indicated in the figure captions.
- e. The culture system in Figure 5 is a cortical-striatal co-culture. That should be indicated in the figure instead of implying that it is a striatal mono-culture.

4. The pharmacological and genetic manipulations of FYN in Figure 6 show preliminary evidence that it plays a critical role in heroin self-administration and cue/drug induced reinstatement.

- a. The heroin self-administration acquisition plots are not shown for the FYN siRNA, did these animals acquire self-administration behavior? It is implied by the increase in active lever presses during the reinstatement tests but should be shown.
- b. It is important to validate that the siRNA FYN manipulations result in knockdown of both mRNA and protein levels. The knockdown in extended Figure 12 is very modest and no protein data is shown. It also appears to be very underpowered at a $n = 2$. The transfection method is questionable and not neuron-specific. It would be better to include a viral-mediated neuron-selective FYN knockdown in these experiments to increase specificity and robustness.
- c. Additionally, there is a disconnect between Figure 5 and 6 in that the FYN manipulations are not connected with any pTau levels. Do pharmacological or siRNA-mediated decreases in FYN prevent the heroin-mediated increases in pTau?
- d. The knockdown/inhibitor experiments were conducted, but there was no overexpression of FYN in neurons to attempt to enhance the heroin-induced increases in pTau.

5. The effects of siRNA and pharmacological inhibition of Fyn on behavior are difficult to interpret.

a. Decreased lever pressing for a single dose of heroin during self-administration is very difficult to interpret. Considering the inverted U-curve for heroin dose-response on lever pressing, decreased lever pressing can be interpreted as either decreased rewarding effects on the ascending limb or as increased rewarding effects on the descending limb of the U-curve. For example, if heroin is more rewarding, then the rat will press less for the same dose of heroin, which is counter to their conclusion. The authors need to determine the effects of Fyn inhibition on dose-response curves for heroin self-administration to disambiguate this issue.

b. As the authors mention regarding their work, Fyn in their studies can be involved in two general processes, including (1) normal glutamatergic signaling and synapse function and (2) some degenerative process similar to that seen in Alzheimer's disease. It is not clear if the authors are inhibiting the normal and necessary Fyn function when they decrease cue-induced reinstatement. Was decreased lever pressing due to decreased levels of degeneration-related Fyn as they imply? Or to decreased levels of the Fyn necessary glutamatergic signaling and synapse function?

c. Comparing the effects of Fyn inhibition on heroin self-administration with food self-administration is a good start. However, the problem is that the cognitive requirements for self-administration of food (same for locomotor activity) are much lower than that for a largely unsensed drug. It would be important to assess the effects of these inhibitors on tasks with higher cognitive requirements.

d. It may be possible to at least partly address the issues in b. and c. by examining the effects of

increasing Fyn (e.g. viral over-expression of Fyn) on behavior.

6. There were some more problems with Figure 5.

a. panel A graph. Normalizing to vehicle artificially reduces the variability of the vehicle control down to zero. The stats should be calculated using raw data.

b. The Western blots here are not helpful. In general, it is not a good idea to show such a restricted area of the blot; this does not indicate the specificity of the immunolabeling or size against molecular weight standards. Thankfully, most people now recommend showing the whole blot in the supplemental and not selected lanes in the main figures. This sample blot thing is just an old bad habit from the old literature.

c. the individual data points should be shown in the graphs in panels B and C.

7. There are many mistakes and inconsistencies in the figures and figure captions. Some are pointed out above, but elements such as labels, text size, and error bars should be closely examined. Plus, many of the graphs need more informative titles that include whether the data come from human, rat or cultured cells; this helps readers avoid having to decode the graphs by going back and forth between the figure and the legend and sometimes the results text.

Reviewer #3 (Remarks to the Author):

Egervari et al performed profiling of chromatin accessibility in neuronal and non-neuronal cells from the striatum of heroin users and matched controls. The authors identified a number of genomic regions with altered chromatin accessibility in heroin users and highlighted the FYN locus as having the most significant differential region. The authors validated that this putative gene regulatory region enhances gene expression and that FYN expression is induced by both heroin and morphine. In addition, the authors showed that FYN, a tyrosine kinase implicated in opioid use and Alzheimer's disease (AD), had increased activity following heroin exposure. Lastly the authors showed that FYN knockdown, and importantly, FYN inhibition using a compound in clinical trials for AD, can attenuate drug seeking behavior in rodents.

The clinical relevance of their findings for the treatment of opioid misuse, which is a huge health crisis in the US, provides a significant impact to the manuscript. However, a major focus of this study is ATAC-seq in postmortem samples of heroin users, which appears to have been previously published (in part) by the authors; unfortunately this is not clearly stated in the main text of the manuscript. There are a number of concerns with the data analysis, which need to be addressed. If these concerns are addressed, it will strengthen the consideration of this manuscript for Nature Communications.

Concerns

1. Information regarding the number of postmortem samples from control and heroin users that were used for the sequencing experiments is listed in Extended Data Table 1. However, these numbers should be clearly stated in the methods and in the results. In addition, the cause of death for the majority of the controls who are in their 20's is listed as 'sudden, natural'; if possible, the authors should provide a better description for the cause of death.

2. ATAC-seq datasets for postmortem samples of heroin users appear to have been previously published for a number of the individuals used in this study, albeit a smaller cohort (comparison of Extended Data Table 1 in the current study and Supplemental Table S1 in Egervari et al 2017 Biol Psychiatry). If this is the case, the authors should clearly state in the main text how many of the

samples used in their analysis were previously published and how many additional samples were added; this should include citation of the previous work and if applicable the accession number of the published data.

3. The authors make the following statement regarding the proportion of variance: 'Most of the variance in chromatin accessibility (average 38.9%) was explained by cell-type (Fig. 2A) Other major contributors included individual variability (average 7.4%), gender (average 3.2%), PMI (average 2.7%) and, importantly, disease (7.5% in neurons and 4.9% in non-neuronal cells).' The authors go on to model disease variance separately for each cell type and subsequently state: 'In this analysis, disease state (i.e. heroin user or control), explained 7.5% and 4.9% of the average variance in neuronal and non-neuronal peaks, respectively. Strikingly, the contribution of disease thus exceeded additional factors such as individual variability (7.4%), gender (3.2%) or PMI (2.7%).' Why are these values in the second analysis the same as in the first analysis? In addition, it appears that the analysis for disease state has already been reported for each cell type in the first analysis and that the analysis for each cell type is not done in the second analysis for individual variability, gender or PMI. This paragraph is confusing and seems repetitive.

4. Figure 2B & C shows heatmaps of 'differential chromatin accessibility analysis for disease status in neuronal and non-neuronal peaks'. How many differential peaks were identified for each cell type? These differential regions were then used for gene ontology analysis in Figure 2D, E. How were chromatin accessibility peaks assigned to genes for this analysis? The authors should also make these gene lists available.

5. Figure 2F, G are volcano plots 'showing differential accessibility regions'; presumably from the text these plots are referring to differential regions between controls and heroin users. However, this is not clearly stated in the legend making this data difficult to interpret. It is also unclear how this analysis differs from that in Figure 2B, C, which also shows differential peaks between controls and heroin users for neurons and non-neurons. Interestingly, the number of differential peaks in Figure 2B, C appears a lot lower than that in Figure F, G; if true, why is this the case?

6. Chromatin accessible regions identified by ATAC-seq are thought to indicate DNA regions of transcription factor binding. The authors should consider analyzing their regions of differential chromatin accessibility between heroin and control groups for enrichment of transcription factor binding motifs. If candidate transcription factors are identified, is there any evidence for gene expression and/or differential gene expression of these transcription factors by RNA-seq.

7. Figure 4A shows that the promoter region of FYN appears more accessible in heroin users compared to controls. Was this region significant in their genome-wide analysis? If not; can the authors quantitatively compare the differential accessibility of the promoter region in neurons (Fig. 4A) versus that of non-neurons (Fig. 4B).

8. Representative western blots of phospho-Fyn and total-Fyn that were used for the quantitation of Fyn activity in Figure 4 should be shown.

9. Figure 5A; the y-axis label reads 'Relative main intensity (normalized to vehicle)'. Should this be relative mean intensity?

10. Representative western blot of GAPDH used as a normalization for quantitating p-Tau levels in Figure 5 should be included.

Response to Reviewers

We thank the Reviewers for their thorough and constructive suggestions. In this revision, we addressed all criticisms and, moreover, include new analyses and experiments that greatly strengthen our study.

The major concerns raised by the Reviewers were with respect to (1) Statistical rigor of the experiments and analysis, (2) Mechanistic role of the putative regulatory region we identified, and (3) Validation of FYN manipulations performed.

Regarding (1), we include new analyses and additional information on the statistical approaches as requested by the Reviewers. These are further explained below in our detailed response.

To provide additional mechanistic insight into the functional role of the putative FYN regulatory region we identified relevant to (2), we include new evidence showing that in neurons, microglia and oligodendrocytes, this region is looped to nearby promoters including that of FYN. Additionally, we show that in the post-mortem human brain, chronic and acute heroin use affects total FYN protein levels.

To further validate the FYN manipulations (3), we performed several additional siRNA injections, and show that these significantly decrease FYN expression in the rat dorsal striatum.

Finally, we addressed all additional technical and conceptual concerns raised by the Reviewers. We re-analyzed data from several experiments as suggested by the Reviewers. Overall, we found that these additional analyses supported our main conclusions. In addition, we have significantly revised the text to ensure that all claims are directly supported by the presented data. As requested, we updated the figures and figure legends to make the study more accessible to the broad readership of *Nature Communications*.

New and revised figures, updated text:

1. New Figures: Ext.D.Fig2e/f, Ext.D.Fig4, Ext.D.Fig5, Ext.D.Fig6, Ext.D.Fig8c-f, Ext.D.Fig14c, Ext.D.Fig17b/c, Ext.D.Fig18, Supplementary Fig 1 and 2
3. Revised Figures: Fig1b, Fig2, Fig5b/c, Ext.D.Fig17a
4. Extensive changes to the main text are highlighted in blue.

Below is our point-by-point response to Reviewers' comments (in blue).

Reviewers' comments:

Reviewer #1 (Remarks to the Author):

This is an elegant and well conducted study that ventured into uncharted territories. The authors should be commended for the breadth of the methods deployed in this study and the originality of the investigations on human brains.

I have to acknowledge that I have limited expertise in epigenetics-related methods so my comments will be more related to other aspects of the study which suffers from several caveats and pitfalls that warrant a thorough revision prior to being considered further for publication.

1. What is the evidence for neurodegeneration-induced protracted cognitive or emotional deficits in heroin addicts? Is it not the case that cognitive deficits substantially decrease following abstinence in former heroin addicts? And what is the evidence that neurodegeneration occurs in the striatum of heroin users or heroin addicts (where the alterations of FYN activity were found in this study)?

Hyperphosphorylated Tau in the post-mortem brain of human heroin users has been previously reported by us (PMID26254956) and other groups (PMIDs 21126996, 10978659, 24292887, 16008828). In our previous study, we observed both an increased percentage of brain regions that contained tau-positive neurites and neurons, as well as increased Tau immunoreactivity in parts of the striatum, such as the caudate nucleus (PMID26254956). In addition, opioid users have been shown to exhibit both transient and, as the Reviewer points out, long-lasting cognitive deficits (e.g. PMIDs 10882838, 24268669, 27649645). In addition, the Reviewer is correct that a direct link between Alzheimer's disease-like pathology, striatal neurodegeneration and cognitive deficits has not yet been established in heroin users. As discussed in the revised manuscript, we do not propose that opioids cause AD, but can lead to general tauopathy. We had emphasized that other signs of AD pathology (amyloid beta depositions) were not detected in heroin users and it is currently unknown whether long-term use leads to striatal neurodegeneration. Intriguingly, a recent study reported neurodegenerative-like mRNA expression profiles in blood samples from over 800 heroin-dependent subjects (PMID29915338). Thus, the observed pTau pathology could potentially represent an early sign of vulnerability to neurodegenerative disorders later in life or reflect a lower end of the Tau-related spectrum related to cognitive impairment. We have revised our discussion to better reflect these important areas of future investigation.

2. Saracatinib is not selective for FYN. FYN is also involved in the trafficking of glutamate receptor subunits and has been shown to be potentially involved in alcohol addiction in humans as well as the reinforcing and incentive properties of cocaine in rats. Can the authors better discuss their findings and their originality in light of the literature on FYN and addiction?

We would like to note that while saracatinib is indeed a non-specific Src kinase family inhibitor, this by no means decreases our enthusiasm for the translational potential of this drug. Indeed, biopharmaceutical innovation is increasingly focused on multispecific drugs (PMID 32296187), especially for disorders with such complex symptoms and etiology as substance use disorders. Going forward, it is of course important to evaluate more selective medications that are suitable for clinical trials, but, to our knowledge, none currently exist. However, saracatinib is well tolerated in preclinical and phase I/II clinical studies and is therefore an excellent candidate for initiating its potential to re-purpose for addiction treatments which are critically needed for opioid addiction.

Regarding the relative contribution of Fyn and other SRC family kinases, we emphasize that none of the other Src family members scored prominently in our variance analysis in the ATAC-seq data. In neurons, the highest ranked ATAC peak associated with SRC was, for example, ranked at #87,400 and only one family member other than FYN had a peak ranked within the top 10,000 (FGR at #8594; Extended Data Fig 10a). In addition, FYN was the only family member altered in the post-mortem brain of the human heroin users, which renders a significant functional contribution of other Src family kinases unlikely. In addition, we observed no changes in Src activity in the dorsal striatum of heroin self-administering animals (Extended Data Fig 10b-d). Finally, siRNA-mediated knock-down of Fyn decreased heroin intake and seeking behaviors (Fig 6), further emphasizing the important role of Fyn among members of the Src kinase family.

The Reviewer is correct that FYN has been linked to several other aspects of substance use disorders including glutamate receptor trafficking, cocaine reinforcement and alcohol self-administration. We have

revised our manuscript to emphasize the previous literature and discuss our findings in light of the aforementioned studies.

3. It is a pity that the legends of fig 1 & 2 (and to some extent those of all the other figures) overall are a compilation of titles for each panel but carry no information or offer no take-home message. This will make the reading of the paper challenging to the broad readership of Nature Communications.

We thank the Reviewer for this suggestion. We have revised all figure legends to improve the description of the data and to make the paper more accessible to broad audiences.

4. In the introduction the authors equate heroin use as a disease (line 1-2). Is this really what they mean? In the revised manuscript, we consistently use substance/heroin use disorder.

5. The dorsal striatum is not involved in reward, is it? Where exactly in the dorsal striatum did the authors measure chromatin accessibility? Clearly the pDMS and the aDLS have opposite functional roles in drug seeking. This should be taken into account and the schematic representation of the brain samples would be much more useful if added to one of the key figures of the paper.

The role of dorsal striatum in mediating goal-directed behaviors, habit formation and compulsive drug taking is well documented (e.g. reviewed by PMID 23438892 or 16715055). Considering the restricted availability of post-mortem human brain samples from heroin addicted individuals and matched controls, our measures of chromatin accessibility were limited to the putamen (Fig 1a). In rats, we focused on the anterior, more medial part of the dorsal striatum, which is the area where long-lasting activation of Fyn has been previously described by Dorit Ron's group during alcohol use (PMIDs 17392475, 20668202, 24588427, 21929909). Nevertheless, it is plausible that different drugs of abuse might affect specific dorsal and ventral striatal regions differently. In future studies, we will aim to further characterize the subregion specificity of our observations both in human and rat. As suggested by the Reviewer, we now depict the schematic representation of brain sampling for chromatin accessibility in the revised Fig1a. We also revised our manuscript to address striatal subregion specificity.

6. Are the epigenetic differences observed between heroin users and controls associated with addiction or instead predating heroin use in the former, and hence represent a factor of vulnerability to, instead of the suggested biological basis of, addiction? Or merely a response to chronic exposure to heroin that is not related to addiction?

We agree that this is an interesting topic that warrants further consideration. Causality, unfortunately, is impossible to assess in our post-mortem human cohort. However, we note that we have previously reported that the epigenetic impairments observed in human are closely mirrored by that observed in heroin self-administering rats. For example, we found hyperacetylation of analogous regions of the GRIA1 gene in human and rat dorsal striatum (PMID 27863698). While the present study did not include ATAC-seq of rat brain, we note that the FYN transcriptional changes were highly similar in humans and rats (compare Fig4c to Fig4d). While we can't rule out this possibility definitively, the fact that the epigenetic/transcriptional impairments are replicated in the animal model suggests that these changes can be induced by heroin, rather than just a potential pre-existing condition that underlies vulnerability.

7. Were the heroin users exclusively using heroin? Did they not also use alcohol or cocaine (which have both been shown to be associated with alterations of Fyn activity?)

Our post-mortem cohort consists of subjects who predominantly used heroin. Persons with documented history or positive toxicology for other abused drugs were excluded. On the other hand, all subjects had a relatively long history of heroin use and died due to heroin overdose. Thus, while we cannot with certainty state that these individuals used heroin exclusively, and while exposure to other substance might have occurred, we made every effort to obtain a homogeneous population and focus on alterations with respect to heroin use. Importantly, blood and urine toxicology at the time of death included assessment of alcohol and we failed to observe any correlations between mRNA/protein levels and blood or urine alcohol concentrations.

8. The authors use interchangeably drug seeking, drug self-administration and reinstatement of an extinguished response for drugs. Clearly each of these processes is mediated by different psychological mechanisms and neither self-administration (which measures reinforcement or taking, as the instrumental response is conflated with the consummatory response under fixed ratio schedules of reinforcement) nor reinstatement (which is the

acquisition of responding for the conditioned reinforcing properties of drug-paired cues under extinction, hence in the absence of the drug, over a single session following extinction) measure drug seeking behaviour. Some clarification is needed in the manuscript as some authors go to great lengths to actually measure drug seeking behaviour or its compulsive nature in preclinical models. I suggest the authors dampen the claims re translational value of these procedures (which have none as far as I understand the psychological mechanisms that subserve the performance of rats in these tasks) or any reference to craving: if lever pressing under extinction is a measure of craving then not giving the lever, hence preventing the animal from responding, is the best way to abolish craving! We should be careful not to overstate what behavioural procedures measure in species that are not endowed with language. Overall the preclinical data are relatively weak: they show that manipulations of Fyn influence the reinforcing properties of heroin.

We completely agree that it is important to refrain from anthropomorphizing animal behavior. That is why we were careful to use “drug self-administration” in relation to drug-taking and did not speak about “liking”. We have diligently gone through the manuscript and did not see the interchangeable use of drug seeking, drug self-administration and reinstatement. Indeed, the drug self-administration behavior and drug seeking measures were discussed in separate sections to be clear about the distinctions.

Not providing a lever does not abolish craving. It abolishes the ability to press a lever previously associated with delivery of the drug. Since rats cannot report “craving”, the field establishes paradigms to provide a proxy of drug seeking where the animal has the ability to press the ‘active lever’ (previously associated with the drug) as compared to pressing the ‘inactive lever’ (not previously associated with the drug) under conditions where no drug is delivered. We therefore used those established models and terminologies. The preclinical data demonstrating that Fyn manipulation *causally* alters heroin self-administration and heroin seeking behavior is very important.

9. Is the decreased level of responding observed at the abstinence test predicted by the overall instrumental output over the last 5/6 sessions of training (influenced by the same manipulations of FYN activity)

This is an interesting point and in fact one of the reasons why we opted for siRNA-mediated knock-down of FYN, given its fast and transient effect on FYN mRNA levels. To avoid potential effects during training, siRNA was injected following the last self-administration session while the animals underwent forced abstinence from heroin. Similarly, saracatinib was only injected after the rats exhibited predetermined acquisition criteria (at least 10 active lever presses and at least 2x as many active as inactive presses on three consecutive days). As suggested by the Reviewer, we tested whether decreased responding during self-administration (during days of saracatinib treatment) was predictive of performance during drug-seeking in the saracatinib cohort. In fact, as shown below in Panel A and Extended Data Fig. 14C of the revised manuscript, we found that active lever pressing during self-administration was significantly correlated with active lever pressing during the heroin seeking test (linear regression; $r=0.67$, $p=0.0124$). This suggests that the saracatinib-induced decrease of active lever pressing during the three self-administration sessions when saracatinib was administered was predictive of the decreased level of responding during the reinstatement test.

Panel A: Active lever pressing during heroin self-administration on days when saracatinib was delivered correlated with active lever pressing during cue-induced drug-seeking behavior.

10. For the acquisition of heroin SA under FR schedule, what is the escalation the authors refer to? Do they mean that the rats displayed a progressive increase in infusions obtained per session which reveals that progressive acquisition of heroin reinforcement at the population level? What is the evidence there is escalation of intake?

We simply refer to the escalation of active lever presses after switching from FR1 to FR5 schedule. Increased lever pressing under these conditions is indicative of increased work effort and thus motivation to obtain heroin. Rats that failed to increase their operant responding during FR5 (compared to FR1) were not included in subsequent experiments. We thank the Reviewer for highlighting this issue. To clarify, we revised the methods section and avoid using 'escalation' throughout the manuscript.

11. For CS-induced reinstatement, (fig 6 A-F), since performance under reinstatement conditions occurs under extinction, the actual influence of the contingent presentation of the CS on responding can only be measured over the first 10 to 20 min of the reinstatement session in comparison with performance over the first 10 to 20 min of the previous extinction session (within-subject design with session as 1 within-subject factor). But the authors did not perform a reinstatement test as the rats had not been exposed to extinction beforehand. Instead they carried out a relapse challenge under extinction. Here the cumulative responses suggest there may be a difference between both groups in their ability to extinguish the influence of the CS over responding (this seems to be a real issue with the siRNA where the scrambled and target siRNA groups show little if no differences in the first 10 min of the reinstatement test). It would be more appropriate to present the actual active and inactive lever presses and carry out an analysis that factors the lever factor in as it seems the drug at the dose used (but clearly not the siRNA) also decreases IL presses (is it a general behavioural suppression?)

We opted for the use of forced abstinence as opposed to extinction training as this paradigm more closely reflects the human condition in which extinction training is never done. The 60 min duration for heroin- or cue-induced drug seeking sessions we employed is standard in the field. Nonetheless, we emphasize that control and saracatinib/siRNA-treated animals behave differently even in the first 10 minutes of these sessions. For example, Fyn siRNA tends to decrease heroin-induced drug seeking ($F_{1,20}=3.136$, $p=0.0918$ from two-way ANOVA). The effects of saracatinib on heroin-induced drug-seeking are highly significant ($F_{1,22}=8.7$, $p=0.0072$ from two-way ANOVA) even within this short time period. Inactive lever presses (or general locomotor activity) are not significantly affected in any of these experiments (e.g. for the saracatinib experiment highlighted by the Reviewer, $F_{1,22}=3.529$, $p=0.0736$ from two-way ANOVA). Taken together, our results suggest that pharmacological or genetic Fyn inhibition attenuates heroin-seeking behavior in rats without impacting general motor behavior. We had presented the actual number of active and inactive lever presses since we agree that is the more appropriate manner of showing the behavioral data.

12. Also, can the authors better justify the use of the dose of saracatinib they used in this study?

We chose the dose of saracatinib based on Kaufman et al. (Ann Neurol 2015, PMID 25707991). Importantly, we demonstrated that at the dose used throughout the study (5 mg/kg), saracatinib significantly decreases active p(Y418)-Fyn and significantly increases inactive p(Y529)-Fyn levels in the dorsal striatum in naïve rats (Extended Data Fig. 10).

13. How do the authors account for the direct effect of morphine on neuronal FYN in cultures? Do the authors suggest these effects are independent of dopamine in the intact striatum? This should be better discussed.

The cell culture findings do indeed suggest that these effects are independent of dopamine, but opioids do not require dopamine for their actions. It would be too speculative to introduce dopamine-specific and dopamine-independent effects from the cell culture model in regard to the behaviors studied *in vivo*. Future studies are needed to fully dissociate dopamine, glutamate and opioid contributions to the effects especially since the contribution of each differs in part at different stages of the addiction cycle.

14. Some titles of the supplementary figures suffer from stylistic issues, for instance one refers to the dose of the drug, but not the drug itself...

We thank the Reviewer for pointing this out. All figure legends have been extensively revised to increase the accessibility of our study to the broad readership of *Nature Communications*.

Reviewer #2 (Remarks to the Author):

This study identifies differential accessible chromatin regions via unbiased ATAC-seq in human post-mortem brain tissue. This analysis revealed a putative regulatory region of the FYN locus, a gene previously implicated in neurodegenerative disorders, as more accessible in a patient population of heroin abusers. Activity of FYN and its downstream target Tau were found to be increased in post-mortem human striatum, rats self-administering heroin, and cultured striatal neurons treated with morphine. Manipulation of FYN via pharmacological and genetic strategies altered self-administration and reinstatement behaviors in a rodent model. Although the variety findings in human post-mortem tissue and model systems very interesting, there are a number of weaknesses described below that need to be addressed before this manuscript can provide a novel, particularly translational finding to the field.

Concerns:

1. The ATAC-seq data does not appear to be rigorously examined and is overall poorly laid out.

a. The QC standards provided in the methods does not indicate if there were any differences between heroin and control samples. The comments below might indicate that there is a technical issue between groups. It is also not indicated in the figure caption how many patient samples derive each sample.

No differences between groups were observed in any of the QC measurements. Figure legends have been updated to reflect number of samples.

b. It is not clear how the adjusted p-values were calculated and if it was corrected for genome-wide multiple comparisons.

In Supplementary Tables 3 and 4, we report nominal p values, F values, logFC and false discovery rate. Different p value and logFC cutoffs were used for different downstream analyses in Figures 1 and 2, which are reported at the appropriate places. Importantly, considering our strict inclusion criteria and the scarce availability of post-mortem brain tissue from predominantly heroin-using individuals, our study was not designed to detect group differences at genome-wide significance. We have revised the title of the manuscript to reflect this. Overall, the cell type is expected to have the largest effect size and given the power of this cohort, we can identify 1000s of peaks with significance at FDR <5%. Not surprisingly, the effect size is much smaller for the heroin vs control comparison, and we fail to see differences after FDR correction. Given the lower power to identify heroin-related peaks at FDR 5%, we decided to select peaks based on the max variance explained by diagnosis.

c. Figure 1a is poorly labeled and should clearly indicate data tracks for both control and heroin samples.

There are no genomic coordinates or scale bars either. The text is very small and is very hard to read.

We revised Fig. 1a to include proper labels, genomic coordinates and scale bars. This panel shows representative tracks from neuronal and non-neuronal cell populations obtained from heroin users and controls.

d. Figure 1b is confusing in that what appears to be the same p-values are represented in 2 different colors. It is also noteworthy that the peaks in the heroin glia also follow the same trend as the neuronal peaks from heroin users in that they too are enriched in CpG islands, promoters, and 5'UTRs. The authors should comment about this and indicate if there is a peak detection issue in the control samples within the dataset.

To clarify the colors of p-values, we added "Enrichment" at the top and "Depletion" at the bottom of the color bar in Figure 1b. Please note that only those peaks unique to heroin or control (for neuron and glia separately) were included in the analysis while the overlap peaks were removed (See the Methods section). The heroin glia peaks are also enriched in the CpG islands, promoters and 5'UTRs with lower significance than heroin neuron. This suggests that heroin increases the accessibility of certain genes in neuron and in glia that are closed in control. The control neuron and glia peaks do not enrich in these genomic regions. This is not due to a detection problem in control samples as most of the peaks are overlapped between heroin and control as shown in the Venn diagrams in Panel B below (revised Extended Data Fig. 2E,F).

Panel B: Overlaps between ATAC peaks in heroin users and controls in neuronal (left) and non-neuronal (right) cell populations

e. Figure 1c is not very clear based on the figure caption. Do the numbers within the heat map correspond to the number of overlapping peaks? Additionally, the publicly available enhancer datasets from ENCODE are from cell lines and are likely to be different between cell culture lines and neurons given cell-type chromatin landscapes, so it is unclear if the overlap is of biological importance. Additionally, if the numbers within the heat maps are the number of peaks overlapping with the specific chromatin mark, then it is alarming that there would be so few overlapping peaks corresponding to active chromatin in the neuron control and that heroin use alone would “suggest that transcriptional activity might be overall increased in neurons of heroin users.” It seems that there are overall more peaks detected in the neuron heroin group from the observation that there are many more overlapping peaks detected across all tested regulatory elements in Figure 1c.

Yes, the numbers in Figure 1c are the number of overlapping peaks. There are indeed more unique peaks (other than the overlapping peaks with control) detected in the heroin neuron group than the other groups. However, the enrichment significance is associated with the proportion of overlapping peaks not the number of overlapping peaks, though more peaks will increase statistical power of Fisher's exact test. Enrichment indicates the proportion of overlapping peaks is significantly larger than the proportion of overlapping by random chance. Therefore, more peaks overlapped does not mean more significantly enriched. For example, the number of overlaps between repressed segments and neuronal peaks in heroin users are much higher than the number of overlaps with peaks in other groups (shown in Figure 1c), while the heroin group is much less significant (even not significant for some cell lines) than the others. Regarding the ENCODE cell line enhancers, they are in fact probably different from normal neurons. But some of those enhancers may be activated by heroin in neurons as the heroin neuron unique chromatin accessible peaks are significantly overlapping with those enhancer regions. There are ~20K protein-coding genes in human genome. Many of them are not normally expressed in neurons at detectable levels but are possibly expressed to response to a heroin stimulus.

f. The text in Figure 2 is overall small and hard to read. Most panels are not labeled in a way that is intuitive to understand.

We increased text size in Fig 2 and revised panel labels. We thank the Reviewer for this suggestion.

g. It is confusing that in Figure 1, the authors conclude that the open chromatin regions in neurons from heroin user implies that there is increased transcriptional activity in those cells. Figure 2 then describes that the less accessible chromatin regions in neurons involve transcription factor activity and RNA polymerase II activity. There is no matching GO-term analysis for the non-neuronal cells even though it is mentioned that non-neuronal regions largely overlapped with neuronal regions.

We agree with the Reviewer and dampened our claims regarding overall transcriptional changes in neurons. While open regions in neurons overlap with enhancer marks and promoters, in lack of cell type-specific RNAseq data from our population, it is not possible to make conclusive statements regarding overall changes in transcriptional activity. Instead, we now simply note that transcriptional activity might be affected based on the ATACseq results. In addition, we performed matching GO analyses from non-neuronal cell populations. The results shown in Panel C (included as Extended Data Fig. 4 in the revised manuscript) highlighted enrichment of categories related to cell adhesion, signal transduction and nucleotide exchange, emphasizing the specificity of our findings.

A

Panel C: Gene ontology analysis of peaks that are less accessible (A) or more accessible (B) in non-neuronal cell populations of heroin users compared to controls.

B

h. It is unclear whether the variance explained in Figure 2 is significant between the different conditions (cell type, disease, gender, etc.). This should be clarified to understand if it is biologically meaningful or random variation. We are not aware of statistical tests to assess whether the amount of variation explained by cell type, disease, etc. is significant. Overall, the cell type is expected to have the largest effect size and given the power of this cohort, we can identify 1000s of peaks with significance at FDR <5%. For disease, the effect size is much smaller and we fail to see differences after FDR correction. Given the lower power to identify disease peaks at FDR 5%, we decided to select peaks based on the max variance explained by diagnosis

2. The variance peak observed ~38kb upstream of the *FYN* locus is not adequately validated as an enhancer of that gene. It should be functionally validated as proximity does not imply that it is an enhancer of the *FYN* gene.

To further corroborate this locus as a putative regulatory element, we queried recently published brain cell type-specific enhancer-promoter interaction maps from the Glass laboratory (Nott et al, Science 2020, PMID 31727856). As shown in Panel D and in the new Extended Figure 6, we found that in neurons, oligodendrocytes and microglia, this locus interacts with both upstream and downstream gene promoters, including that of *FYN*. This, together with the luciferase data presented in Figure 3, strongly supports a putative enhancer role of this region of interest in regulating the expression of *FYN* gene. In addition, we decided to tone down our conclusions and, as explained below, we now use “putative” regulatory element throughout the revised manuscript.

Panel D: PLAC-seq data from Nott et al, Science 2020 indicate that in neurons, oligodendrocytes and microglia, this locus interacts with both upstream and downstream gene promoters, including that of FYN.

- a. The raw ATAC signal shown in Figure 3a largely looks similar between control and heroin samples, but the marginal increase appears to be mostly in the control group over the heroin group. We emphasize that this region was identified based on the variance analysis and not based on group difference. While the exact regulatory role of this putative element remains to be further investigated, we show extensive follow-up data that describes increased expression and activity of the Fyn kinase, as well as its functional importance for addiction-related behaviors.
- b. The luciferase assay indicates that the region can drive increased gene expression. This could be the result of having an expanded promoter in the luciferase plasmid as there was not a control sequence (scrambled or otherwise) inserted as a control to correct for increased promoter length. We maintain that the result of the luciferase assay supports the potential enhancer function of this region of interest. We note that promoter length is not affected by the insert and that using empty expression vector as a control is by not unusual in this type of experiment. Nevertheless, we were in the process of performing the requested control and cloned a scrambled sequence into the pGL3-promoter vector. Unfortunately, due to an unexpected delay in obtaining the scrambled sequence, and then the COVID-19 pandemic, we were not able to complete these experiments. Therefore, we decided to tone down our conclusions and, as explained below, we use “putative” regulatory element throughout the revised manuscript. We thank the Reviewers and the Editors for their understanding.

c. Additionally, increased luciferase activity does not indicate that the region is a definitive enhancer of the FYN locus. To establish this locus as an enhancer, it should be functionally tested with a deletion or inactivation, even if only in a cell line.

The revised manuscript includes evidence for looping between the putative regulatory region and FYN promoter. Together, these results suggest that this locus might function as a regulatory element that affects FYN transcription. Nevertheless, we agree with the Reviewer that we did not definitively establish this locus as a FYN enhancer. Therefore, throughout the revised manuscript, we refer to this region as a putative regulatory element.

3. Figures 4 and 5 present interesting observations regarding the levels of FYN mRNA and protein in a variety of model systems.

a. Figure 4a displays the promoter of FYN over the putative enhancer in Figure 3. There appears to be an increase in raw signal between control and heroin neuron samples, but this is not indicated as significant and could be the case if more peaks were detected in the neuron heroin group as mentioned in a previous concern. As suggested by Reviewer 2 and Reviewer 3, we calculated the average ATAC signal over this region. In non-neuronal cells, we observed a 1.3-fold increase (two-tailed Student's t test $p=2.3 \times 10^{-10}$). As shown in Fig. 4A, the difference in accessibility was more substantial in neuronal cells with a 2.2-fold increase in heroin users compared to controls ($p=7.64 \times 10^{-93}$). These results are now included in the revised manuscript.

b. The data in Figure 4f-q appears to be normalized to the saline group, which should be clarified. It is not indicated if total levels of FYN protein are altered based on the way the data is normalized. If there is increased transcription and enhancer activity, it would be important to measure and display the levels of total protein. Also, the authors should comment on how potentially increasing the protein itself also results in more activated protein.

We revised the figure legend to indicate normalization to saline. As suggested by the Reviewer, we measured Total FYN protein levels in the post-mortem human putamen of heroin users and matched controls. Interestingly, while we did not observe overall group differences, FYN protein showed significant correlations with specific demographic variables in subjects where this information was available to us. Namely, just as for FYN mRNA (Extended Data Fig 8), we found a positive correlation with years of previous drug use and a negative correlation with urine morphine levels at the time of death. Overall, these data suggest that chronic heroin use increases total levels of FYN in the human putamen, while acute drug exposure has an antagonistic effect. This is intriguing as we previously reported similar findings for histone acetylation in a partially overlapping cohort of post-mortem human heroin users (Egervari et al, Biol Psych 2017), suggesting differential effects of chronic heroin-related pathology and acute drug toxicity. We thank the Reviewer for this suggestion and include the new findings in the revised manuscript as Extended Data Fig 8C-E.

Panel E: Total FYN protein in the post-mortem putamen of human heroin users. A. Western blots revealed no significant group differences. B. Total FYN protein showed a negative correlation with urine morphine at time of death. C. Total FYN protein showed a positive correlation with years of previous drug use.

In further support, total Fyn protein levels were increased in a seemingly dose-dependent manner in striatal monocultures chronically treated with morphine. This is now included as Extended Data Fig 8F.

FYN in primary striatal monocultures

Panel F: In primary striatal monocultures chronically treated with morphine, total Fyn protein levels show a dose dependent increase.

c. Though it may not take away from the experimental interpretation, there were multiple schedules of self-administration sessions described in the methods for Figure 4. It is unclear why.

The animals used in Figure 4F-J underwent a slightly different self-administration paradigm for external reasons (tissue was used from an older cohort of rats). The same inclusion criteria were used as in other cohorts and this difference did not affect the interpretation of the data.

d. Replicates are not indicated in the figure captions.

Figure legends have been revised to indicate number of replicates.

e. The culture system in Figure 5 is a cortical-striatal co-culture. That should be indicated in the figure instead of implying that it is a striatal mono-culture.

The figure legend has been revised to indicate cortical-striatal co-culture.

4. The pharmacological and genetic manipulations of FYN in Figure 6 show preliminary evidence that it plays a critical role in heroin self-administration and cue/drug induced reinstatement.

a. The heroin self-administration acquisition plots are not shown for the FYN siRNA, did these animals acquire self-administration behavior? It is implied by the increase in active lever presses during the reinstatement tests but should be shown.

We used the same predetermined acquisition criteria in all self-administration experiments (at least 10 active lever presses, and at least 2x as many active as inactive presses on three consecutive days). Importantly, to avoid a potential effect on acquisition in these experiments, FYN siRNA was delivered after the animals acquired self-administration. Active and inactive lever presses for these cohorts are now included as Extended Data Fig 18 in the revised manuscript.

b. It is important to validate that the siRNA FYN manipulations result in knockdown of both mRNA and protein levels. The knockdown in extended Figure 12 is very modest and no protein data is shown. It also appears to be very underpowered at a $n = 2$. The transfection method is questionable and not neuron-specific. It would be better to include a viral-mediated neuron-selective FYN knockdown in these experiments to increase specificity and robustness.

We performed additional validation of the siRNA manipulations by assessing knock-down efficiency in additional naïve Long-Evans rats ($n=7$). siRNA was infused in the right hemisphere and control siRNA in the left hemisphere. As shown in Panel G, 3 days following the injection of 20 pmol siRNA, we observed a significant reduction of FYN mRNA in the dorsal striatum (paired Student's t-test, $p=0.008$). Extended Data Figure 17 has been updated to include the new data. Due to the COVID-19 lockdown, we were not able to determine protein levels. Nonetheless, the mRNA data suggest that we achieved an effective knock-down of FYN and the behavioral effects replicate the findings with saracatinib.

Fyn 20 pmol siRNA 3 days

Panel G: Injection of 20 pmol Fyn siRNA into rat caudate-putamen results in significantly decreased Fyn mRNA levels 3 days post-injection.

c. Additionally, there is a disconnect between Figure 5 and 6 in that the FYN manipulations are not connected with any pTau levels. Do pharmacological or siRNA-mediated decreases in FYN prevent the heroin-mediated increases in pTau?

Due to the short and transient nature of our saracatinib treatment (maximum of 3 injections), we did not expect a complete reversal in heroin-mediated increases of phosphorylated Tau. Nevertheless, we observed a trend for decreased Tau phosphorylation in saracatinib treated rats (Extended Data Fig 15b). This, together with decreased kinase activity (Extended Data Fig 15a) and the positive relationship between dorsal striatal protein tyrosine kinase activity and pTau levels (Extended Data Fig 15c) suggest that heroin-induced FYN activity plays a role in striatal Tau phosphorylation, which could be attenuated by pharmacological or siRNA-mediated inhibition of FYN. We are testing this possibility in follow-up studies.

d. The knockdown/inhibitor experiments were conducted, but there was no overexpression of FYN in neurons to attempt to enhance the heroin-induced increases in pTau.

We focused on pharmacological (and genetic) FYN inhibition due to its potential translational and therapeutic relevance. Nonetheless, we agree with the Reviewer that FYN overexpression is of potential interest to further corroborate the role of this enzyme in regulating heroin-induced Tau phosphorylation and behavioral phenotypes. We now mention this in the revised discussion.

5. The effects of siRNA and pharmacological inhibition of Fyn on behavior are difficult to interpret.

a. Decreased lever pressing for a single dose of heroin during self-administration is very difficult to interpret. Considering the inverted U-curve for heroin dose-response on lever pressing, decreased lever pressing can be interpreted as either decreased rewarding effects on the ascending limb or as increased rewarding effects on the descending limb of the U-curve. For example, if heroin is more rewarding, then the rat will press less for the same dose of heroin, which is counter to their conclusion. The authors need to determine the effects of Fyn inhibition on dose-response curves for heroin self-administration to disambiguate this issue.

We respectfully disagree with the statement that dose-response curves of heroin self-administration are necessary in this particular study. We use a standard dose of heroin that is translationally relevant and that has been used extensively in the literature by our group and others. Most importantly, our conclusions do not solely rely on self-administration data. We have strong evidence using orthogonal approaches that demonstrates that pharmacological (saracatinib) and genetic (Fyn siRNA) inhibition of Fyn decreases drug- and cue-induced heroin-seeking behavior. These are state of the art and highly translational behavioral paradigms that outline Fyn kinase as a potential therapeutic target for heroin use disorder.

b. As the authors mention regarding their work, Fyn in their studies can be involved in two general processes,

including (1) normal glutamatergic signaling and synapse function and (2) some degenerative process similar to that seen in Alzheimer's disease. It is not clear if the authors are inhibiting the normal and necessary Fyn function when they decrease cue-induced reinstatement. Was decreased lever pressing due to decreased levels of degeneration-related Fyn as they imply? Or to decreased levels of the Fyn necessary glutamatergic signaling and synapse function?

We show that Fyn mRNA levels and Fyn kinase activity are increased by chronic exposure to heroin. In addition, we show that saracatinib has no effect on food self-administration and other general behavioral outcomes, for example locomotor activity. Thus, our data suggest that the saracatinib treatment inhibits pathological, heroin-induced functions of Fyn, rather than its normal and necessary activity. Nevertheless, we agree with the Reviewer that, considering the role of Fyn in glutamatergic synaptic function, this is an interesting topic for further investigation.

c. Comparing the effects of Fyn inhibition on heroin self-administration with food self-administration is a good start. However, the problem is that the cognitive requirements for self-administration of food (same for locomotor activity) are much lower than that for a largely unsensed drug. It would be important to assess the effects of these inhibitors on tasks with higher cognitive requirements.

It is unclear which tasks the Reviewer is referring to. However, food self-administration is the general standard to help clarify the specificity of manipulations relevant to general reward.

d. It may be possible to at least partly address the issues in b. and c. by examining the effects of increasing Fyn (e.g. viral over-expression of Fyn) on behavior.

As mentioned above, we focused on pharmacological (and genetic) FYN inhibition due to its potential translational and therapeutic relevance. Nonetheless, we agree with the Reviewer that FYN overexpression is of potential interest to further corroborate the role of this enzyme in regulating heroin-induced Tau phosphorylation and behavioral phenotypes. We now mention this in the revised discussion.

6. There were some more problems with Figure 5.

a. panel A graph. Normalizing to vehicle artificially reduces the variability of the vehicle control down to zero. The stats should be calculated using raw data.

As expected with primary cell cultures, there is variability between cell preparations which would make the un-normalized data difficult to interpret. Normalizing to vehicle is standard procedure in these assays to compare data from different batches. Nonetheless, we re-analyzed the data as requested by the Reviewer. Of note, within each replicate, there is either a trend or a statistically significant increase of Tau phosphorylation (see Panel H), supporting our original conclusions.

Panel H: Immunohistochemistry of 3 independent biological replicates revealed increased Tau phosphorylation in morphine-treated primary striatal cell cultures. These data were not normalized to vehicle controls.

b. The Western blots here are not helpful. In general, it is not a good idea to show such a restricted area of the blot; this does not indicate the specificity of the immunolabeling or size against molecular weight standards. Thankfully, most people now recommend showing the whole blot in the supplemental and not selected lanes in the main figures. This sample blot thing is just an old bad habit from the old literature.

We agree and include all uncropped Western blots as Supplementary Figures 1 and 2 in the revised manuscript.

c. the individual data points should be shown in the graphs in panels B and C.

Fig. 5B/C has been revised to show individual data points in these panels.

7. There are many mistakes and inconsistencies in the figures and figure captions. Some are pointed out above, but elements such as labels, text size, and error bars should be closely examined. Plus, many of the graphs need more informative titles that include whether the data come from human, rat or cultured cells; this helps readers avoid having to decode the graphs by going back and forth between the figure and the legend and sometimes the results text.

We thank the Reviewer for this suggestion. Figure legends, graph titles and additional elements have been revised to increase clarity and to make the paper more accessible to the readers of *Nature Communications*.

Reviewer #3 (Remarks to the Author):

Egervari et al performed profiling of chromatin accessibility in neuronal and non-neuronal cells from the striatum of heroin users and matched controls. The authors identified a number of genomic regions with altered chromatin accessibility in heroin users and highlighted the FYN locus as having the most significant differential region. The authors validated that this putative gene regulatory region enhances gene expression and that FYN expression is induced by both heroin and morphine. In addition, the authors showed that FYN, a tyrosine kinase implicated in opioid use and Alzheimer's disease (AD), had increased activity following heroin exposure. Lastly the authors showed that FYN knockdown, and importantly, FYN inhibition using a compound in clinical trials for AD, can attenuate drug seeking behavior in rodents.

The clinical relevance of their findings for the treatment of opioid misuse, which is a huge health crisis in the US, provides a significant impact to the manuscript. However, a major focus of this study is ATAC-seq in postmortem samples of heroin users, which appears to have been previously published (in part) by the authors; unfortunately this is not clearly stated in the main text of the manuscript. There are a number of concerns with the data analysis, which need to be addressed. If these concerns are addressed, it will strengthen the consideration of this manuscript for *Nature Communications*.

Concerns

1. Information regarding the number of postmortem samples from control and heroin users that were used for the sequencing experiments is listed in Extended Data Table 1. However, these numbers should be clearly stated in the methods and in the results. In addition, the cause of death for the majority of the controls who are in their 20's is listed as 'sudden, natural'; if possible, the authors should provide a better description for the cause of death.

We updated the results and methods sections to include the number of postmortem samples from control and heroin users used in ATACseq and qPCR experiments. In addition, we revised Extended Data Table 1 to reflect more detailed information about cause of death for all control subjects where that information is available.

2. ATAC-seq datasets for postmortem samples of heroin users appear to have been previously published for a number of the individuals used in this study, albeit a smaller cohort (comparison of Extended Data Table 1 in the current study and Supplemental Table S1 in Egervari et al 2017 *Biol Psychiatry*). If this is the case, the authors should clearly state in the main text how many of the samples used in their analysis were previously published and how many additional samples were added; this should include citation of the previous work and if applicable the accession number of the published data.

The Reviewer is correct that we have included individual ATACseq tracks from this experiment in our previous publication to show differential accessibility of the *GRIA1* gene body in heroin users vs controls (see Fig. 3 of Egervari et al, Biol Psych 2017). However, the ATACseq data and its detailed analysis have not been previously published and will be made accessible upon publication of this manuscript. To our knowledge, no such dataset currently exists, which we believe will make it an invaluable resource for the scientific community.

3. The authors make the following statement regarding the proportion of variance: 'Most of the variance in chromatin accessibility (average 38.9%) was explained by cell-type (Fig. 2A) Other major contributors included individual variability (average 7.4%), gender (average 3.2%), PMI (average 2.7%) and, importantly, disease (7.5% in neurons and 4.9% in non-neuronal cells).' The authors go on to model disease variance separately for each cell type and subsequently state: 'In this analysis, disease state (i.e. heroin user or control), explained 7.5% and 4.9% of the average variance in neuronal and non-neuronal peaks, respectively. Strikingly, the contribution of disease thus exceeded additional factors such as individual variability (7.4%), gender (3.2%) or PMI (2.7%).' Why are these values in the second analysis the same as in the first analysis? In addition, it appears that the analysis for disease state has already been reported for each cell type in the first analysis and that the analysis for each cell type is not done in the second analysis for individual variability, gender or PMI. This paragraph is confusing and seems repetitive.

We thank the Reviewer for pointing this out. Cell type-specific analysis was only performed for disease state. We revised this paragraph to clarify and to avoid repetition.

4. Figure 2B & C shows heatmaps of 'differential chromatin accessibility analysis for disease status in neuronal and non-neuronal peaks'. How many differential peaks were identified for each cell type? These differential regions were then used for gene ontology analysis in Figure 2D, E. How were chromatin accessibility peaks assigned to genes for this analysis? The authors should also make these gene lists available.

Using a cutoff of nominal p value < 0.01 in the heroin vs control comparison, we identified 729 differentially accessible peaks in neurons and 1111 differentially accessible peaks in non-neuronal cells. 51 and 57 of these had a $|\logFC| > 1.7$ (shown in Fig2B/C). Peaks were assigned to genes based on the nearest transcriptional start site. All neuronal and non-neuronal peaks are listed in Extended Data Tables 3 and 4, respectively. Nominal p values and logFCs are included in these tables.

5. Figure 2F, G are volcano plots 'showing differential accessibility regions'; presumably from the text these plots are referring to differential regions between controls and heroin users. However, this is not clearly stated in the legend making this data difficult to interpret. It is also unclear how this analysis differs from that in Figure 2B, C, which also shows differential peaks between controls and heroin users for neurons and non-neurons. Interestingly, the number of differential peaks in Figure 2B, C appears a lot lower than that in Figure F, G; if true, why is this the case?

The Reviewer is correct that differential accessibility refers to the heroin vs control comparison; we revised the figure legends to clarify this. With respect to the number of peaks, for the volcano plots in Fig 2F/G, we defined differentially accessible regions using a cutoff of nominal p value < 0.01. For the heatmaps in Fig 2B/C, we further filtered the list and only included peaks with a nominal p value < 0.01 and $|\logFC| > 1.7$.

6. Chromatin accessible regions identified by ATAC-seq are thought to indicate DNA regions of transcription factor binding. The authors should consider analyzing their regions of differential chromatin accessibility between heroin and control groups for enrichment of transcription factor binding motifs. If candidate transcription factors are identified, is there any evidence for gene expression and/or differential gene expression of these transcription factors by RNA-seq.

We thank the Reviewer for this suggestion. In response, we determined DNA binding factor occupancy genome wide by performing footprinting analysis of our neuronal ATACseq data as previously described (PMID 24097267). As shown in Panel I and the new Extended Data Figure 5, we found significant enrichment of several transcription factors and DNA-binding proteins such as DNMT1 or SMAD1 in heroin users compared to controls. While we do not have RNAseq data from our post-mortem population at this time, we note that several transcription factors with higher occupancy in heroin users were strongly implicated in neuronal function, behavior and addiction. These included, for example, MBD2 (PMID27303433, PMID22438930) and MeCP2 (PMID20711185; PMID20711186).

Panel I: ATACseq footprinting analysis reveals significant enrichment of transcription factors and DNA binding proteins in the post-mortem human putamen of human heroin users compared to matched controls.

7. Figure 4A shows that the promoter region of FYN appears more accessible in heroin users compared to controls. Was this region significant in their genome-wide analysis? If not; can the authors quantitatively compare the differential accessibility of the promoter region in neurons (Fig. 4A) versus that of non-neurons (Fig. 4B).

This peak was not identified as differentially accessible in our genome-wide analysis. As suggested by the Reviewer, we calculated the average ATAC signal over this region. In non-neuronal cells, we observed a 1.3-fold increase (two-tailed Student's t test $p=2.3 \times 10^{-10}$). As shown in Fig. 4A, the difference in accessibility was more substantial in neuronal cells with a 2.2-fold increase in heroin users compared to controls ($p=7.64 \times 10^{-93}$). These results are now included in the revised manuscript.

8. Representative western blots of phospho-Fyn and total-Fyn that were used for the quantitation of Fyn activity in Figure 4 should be shown.

We now show all uncropped pTau and total Fyn Western blots in Supplementary Figures 1 and 2.

9. Figure 5A; the y-axis label reads 'Relative main intensity (normalized to vehicle)'. Should this be relative mean intensity?

We corrected the label and thank the Reviewer for pointing this out.

10. Representative western blot of GAPDH used as a normalization for quantitating p-Tau levels in Figure 5 should be included.

We now show all uncropped p-Tau and GAPDH Western blots in Supplementary Figures 1 and 2.

REVIEWERS' COMMENTS:

Reviewer #1 (Remarks to the Author):

General comments:

The authors have done an excellent job addressing my previous concerns and that of the other reviewers.

I would like to point out that the demonstration that siRNA-mediated downregulation of the target mRNA levels coinciding with clear behavioural effects is sufficient, without an assessment of protein levels, to draw firm conclusions as to the causal nature of that effect in the hypothesis-driven framework within which the experiment was designed. Similarly, I do not understand the rationale for suggesting that the heroic experiment performed here, with converging evidence from studies in humans (this is the real-life picture of the disease!!) and animals, the latter using a dose of heroin which precludes interpretations of shifts in the reinforcing or satiety effect of the drug that may account for the observed results.

I learnt a lot in the process and I strongly recommend the paper be considered for publication pending only a couple of minor amendments that will likely not require re-review.

Minor points:

I would like to thank the authors for their engagement in a constructive discussion over the comments of the reviewers.

There are a couple of outstanding issues that I would be grateful the authors paid attention to:

1. I am sorry to be insisting on that aspect, but... Line 236-237 : the authors refer to models of heroin use disorder.... But, self-administration is the operationalisation of drug reinforcement, and it is a model of a short history of controlled recreational, drug use, so whatever behaviour is measured following a short exposure to heroin self-administration, it simply cannot be said to measure HUD. It has long been recognised by the field that self-administration is not a model of addiction or drug use disorder. Similarly, measuring responding under extinction is not similar to measuring drug seeking (line 324 and throughout the ms): here the animals were challenged to acquire responding for the conditioned reinforcing properties of the drug-paired cue, but under extinction.

Drug seeking is responding for, but in the absence of, the drug, eventually resulting in its procurement and consumption. Thus, drug seeking does not display a time-dependent decay in responding as observed under extinction conditions (as used by the authors here). Under extinction the organism learns a new response-non US association and never receives the drug. Quantitative differences in instrumental responding can be due to the motivational state of the animal, previous propensity instrumentally to respond (as the authors show themselves here), and/or learning abilities.

The field used to agree on the fact that IVSA was the goal-standard model of addiction, they the field agreed that extinction-reinstatement was the goal-standard model of addiction (relapse/craving), it is not because the field mis-understands responding under extinction over a single session as drug seeking that we all should abide by this dogma.

I suggest the authors remain procedural in their description of the previous data to which they refer, or the tasks they used here: either their target system is involved in heroin reinforcement, in responding under extinction in a relapse test following forced abstinence, or extinction.

2. Interestingly, the 329 saracatinib-induced decrease of active lever pressing during the three self-administration 330 sessions when saracatinib was administered was predictive of the decreased level of 14 331 responding during the subsequent drug-seeking test ($n=13$, $r=0.67$, $p=0.0124$ from linear 332 regression; Extended Data Fig. 14C).

I thank the authors for their openness to the previous comments. However, their description that performance at test under extinction following differential treatment initiated during SA is

predicted by instrumental performance during SA warrants a re-appraisal of the interpretation of the relapse-related results: the behavioural manifestations at relapse are seemingly due to a carry-over effect of behavioural adaptations that were induced during reinforcement. A measurement of the effect of treatment on the persistence of responding at relapse would require the introduction of treatment during abstinence. I am not suggesting the authors should do such experiment here, and I do not think the conclusions of the paper would be strengthened by such experiment. However, I encourage the authors to slightly revisit their interpretation and discussion of this result.

Reviewer #2 (Remarks to the Author):

Overall, the revised manuscript addresses many of my initial concerns. The following are minor concerns/suggestions, mostly to expand the discussion.

1-The limitations of the statistics available for this cohort should be emphasized since the authors mentioned that the study was not designed to detect group differences at a genome-wide significance level.

2-Include rationale about why the non-overlapping peaks were selected for the analysis in Figure 1b and perhaps a flow chart to show the analysis pipeline. What do you see if you include them? Are the non-overlapping peaks associated with neuron and glia-specific genes? This might be a nice addition to the figure.

3-It is understandable that the control luciferase experiment and the protein expression experiment could not be completed, and they should not hold up publication of this manuscript.

4-The author's comments (response to comment 4c) about the relationship between saracatinib treatment and Tau phosphorylation should be added to the discussion.

5-While the figure caption updates are great, some of the text in the figures is still on the small side, such as Figure 2b-c, 2f-g, 3a-b. Overall, the figure panels are not consistent in font sizes and would look more polished if they were more consistent within and between figures.

Minor comment: some text in figures is still hard to read or is overlapping, please check.

Reviewer #3 (Remarks to the Author):

The revised manuscript of Egervari et al has provided new data and analysis, as well as substantial edits to the main text, legends and figures. This has strengthened the conclusions of the manuscript and has provided further clarity that will aid interpretation of their findings by the broad audience of Nature Communications. In addition, the authors have provided the reviewers full-length versions of their western blots as Supplementary Figures, rather than as Extended Data Figures. It is unclear whether these Supplementary Figures 1 and 2 are for the reviewers only or they will be published. Regardless, the authors should include these full-length western blots in their final publication. Besides this one minor point, the revised manuscript is of sufficient interest and rigor for publication in Nature Communications.

REVIEWERS' COMMENTS:

We thank the Reviewers for their positive assessment of the revised manuscript. As detailed below in our point by point response, we addressed all remaining minor points from the Reviewers.

Reviewer #1 (Remarks to the Author):

General comments:

The authors have done an excellent job addressing my previous concerns and that of the other reviewers.

I would like to point out that the demonstration that siRNA-mediated downregulation of the target mRNA levels coinciding with clear behavioural effects is sufficient, without an assessment of protein levels, to draw firm conclusions as to the causal nature of that effect in the hypothesis-driven framework within which the experiment was designed. Similarly, I do not understand the rationale for suggesting that the heroic experiment performed here, with converging evidence from studies in humans (this is the real-life picture of the disease!!) and animals, the latter using a dose of heroin which precludes interpretations of shifts in the reinforcing or satiety effect of the drug that may account for the observed results.

I learnt a lot in the process and I strongly recommend the paper be considered for publication pending only a couple of minor amendments that will likely not require re-review.

Minor points:

I would like to thank the authors for their engagement in a constructive discussion over the comments of the reviewers.

There are a couple of outstanding issues that I would be grateful the authors paid attention to:

1. I am sorry to be insisting on that aspect, but... Line 236-237 : the authors refer to models of heroin use disorder.... But, self-administration is the operationalisation of drug reinforcement, and it is a model of a short history of controlled recreational, drug use, so whatever behaviour is measured following a short exposure to heroin self-administration, it simply cannot be said to measure HUD. It has long been recognised by the field that self-administration is not a model of addiction or drug use disorder. Similarly, measuring responding under extinction is not similar to measuring drug seeking (line 324 and throughout the ms): here the animals were challenged to acquire responding for the conditioned reinforcing properties of the drug-paired cue, but under extinction.

Drug seeking is responding for, but in the absence of, the drug, eventually resulting in its procurement and consumption. Thus, drug seeking does not display a time-dependent decay in responding as observed under extinction conditions (as used by the authors here). Under extinction the organism learns a new response-non US association and never receives the drug. Quantitative differences in instrumental responding can be due to the motivational state of the animal, previous propensity instrumentally to respond (as the authors show themselves here), and/or learning abilities.

The field used to agree on the fact that IVSA was the goal-standard model of addiction, they the field agreed that extinction-reinstatement was the goal-standard model of addiction (relapse/craving), it is not because the field mis-understands responding under extinction over a single session as drug seeking that we all should abide by this dogma.

I suggest the authors remain procedural in their description of the previous data to which they refer, or the tasks they used here: either their target system is involved in heroin reinforcement, in responding under extinction in a relapse test following forced abstinence, or extinction.

We agree that the adjudication of “drug-seeking” behavior is limited in preclinical models, as they do not assess subsequent procurement of drug after elevated motivation in responding. Despite these

shortcomings, drug self-administration and responding for drug-paired cues do recruit similar neurobiological mechanisms and have relevance to the human condition, and still serve as a widely used rodent model for addiction-like behaviors. However, we have changed all “drug-seeking” statements to “cue-induced responding”, or “active lever responding” to be more procedural in our language, and have stated that these behaviors are “akin to drug-seeking” to describe their potential relevance to human substance abuse. In addition, we added the following statement to the discussion: “future studies are required to further delineate the role of Fyn inhibition on more nuanced aspects of these behaviors (e.g. habitual versus goal oriented responding for drug or cues).” The purpose of the current in vivo studies was to validate a causal role in Fyn’s contribution to heroin self-administration and cue-induced responding, which has been demonstrated.

2. Interestingly, the 329 saracatinib-induced decrease of active lever pressing during the three self-administration 330 sessions when saracatinib was administered was predictive of the decreased level of 14 331 responding during the subsequent drug-seeking test (n=13, r=0.67, p=0.0124 from linear 332 regression; Extended Data Fig. 14C).

I thank the authors for their openness to the previous comments. However, their description that performance at test under extinction following differential treatment initiated during SA is predicted by instrumental performance during SA warrants a re-appraisal of the interpretation of the relapse-related results: the behavioural manifestations at relapse are seemingly due to a carry-over effect of behavioural adaptations that were induced during reinforcement. A measurement of the effect of treatment on the persistence of responding at relapse would require the introduction of treatment during abstinence. I am not suggesting the authors should do such experiment here, and I do not think the conclusions of the paper would be strengthened by such experiment. However, I encourage the authors to slightly revisit their interpretation and discussion of this result.

We believe that the decrease in cue-induced responding is not simply due to carry-over effects, as there was no carry-over effect from the saracatinib self-administration sessions to the subsequent saracatinib-free self-administration session (shown in Fig 6A). However, we agree that the possibility that the cue-induced reinstatement effect may be due to changes in operant responding because of previous experience with saracatinib cannot be eliminated based on the present data. Therefore, we included the following statement in the revised manuscript: “future experiments where saracatinib is only introduced during abstinence are warranted to confirm that the drug has specific effects to cue-induced reinstatement versus long-term depression of operant responding”.

Reviewer #2 (Remarks to the Author):

Overall, the revised manuscript addresses many of my initial concerns. The following are minor concerns/suggestions, mostly to expand the discussion.

1-The limitations of the statistics available for this cohort should be emphasized since the authors mentioned that the study was not designed to detect group differences at a genome-wide significance level.

We added the following statement to the manuscript: “This analysis is of particular interest for smaller post-mortem human cohorts, where criteria for genome-wide significance are rarely met.”

2-Include rationale about why the non-overlapping peaks were selected for the analysis in Figure 1b and perhaps a flow chart to show the analysis pipeline. What do you see if you include them? Are the non-overlapping peaks associated with neuron and glia-specific genes? This might be a nice addition to the figure.

Since our primary interest was to assess how heroin affects chromatin accessibility, we focused our analysis on non-overlapping peaks. Shared peaks between heroin users and controls are related to homeostatic genes, the inclusion of which could obscure downstream analyses. This is now indicated in the revised results section. The detailed analytic pipeline is included in the Methods section under 'ATAC-seq data processing'.

3-It is understandable that the control luciferase experiment and the protein expression experiment could not be completed, and they should not hold up publication of this manuscript.

We thank the Reviewers for their understanding.

4-The author's comments (response to comment 4c) about the relationship between saracatinib treatment and Tau phosphorylation should be added to the discussion.

We added the following statement to the discussion: "Intriguingly, saracatinib attenuated neurodegenerative-like pathology in rats, although no complete reversal was observed due to the short duration of treatment". In addition, we note that this point is also addressed in the results section: "These effects did not reach statistical significance, likely due to the non-specificity of the PTK assay used and the relatively short duration of treatment."

5-While the figure caption updates are great, some of the text in the figures is still on the small side, such as Figure 2b-c, 2f-g, 3a-b. Overall, the figure panels are not consistent in font sizes and would look more polished if they were more consistent within and between figures.

We increased the text size in the aforementioned panels and aimed to eliminate inconsistencies in fonts.

Minor comment: some text in figures is still hard to read or is overlapping, please check.

We checked all figures to ensure that there is no overlapping or otherwise unreadable text.

Reviewer #3 (Remarks to the Author):

The revised manuscript of Egervari et al has provided new data and analysis, as well as substantial edits to the main text, legends and figures. This has strengthened the conclusions of the manuscript and has provided further clarity that will aid interpretation of their findings by the broad audience of Nature Communications. In addition, the authors have provided the reviewers full-length versions of their western blots as Supplementary Figures, rather than as Extended Data Figures. It is unclear whether these Supplementary Figures 1 and 2 are for the reviewers only or they will be published. Regardless, the authors should include these full-length western blots in their final publication. Besides this one minor point, the revised manuscript is of sufficient interest and rigor for publication in Nature Communications.

The full-length Western blots will be published and available to readers in the Supplementary Information.